# The splicing regulators Esrp1 and Esrp2 direct an epithelial splicing program essential for mammalian development

**Thomas W Bebee[1†], Juw Won Park[2†‡], Katherine I Sheridan[1], Claude C Warzecha[1], Benjamin W Cieply[1], Alex M Rohacek[3], Yi Xing[2]\*, Russ P Carstens[1,3]\***

[1]Department of Medicine, Perelman School of Medicine, University of Pennsylvania, Philadelphia, United States; [2]Department of Microbiology, Immunology and Molecular Genetics, University of California, Los Angeles, Los Angeles, United States; [3]Department of Genetics, Perelman School of Medicine, University of Pennsylvania, Philadelphia, United States

**Abstract** Tissue- and cell-type-specific regulators of alternative splicing (AS) are essential components of posttranscriptional gene regulation, necessary for normal cellular function, patterning, and development. Mice with ablation of Epithelial splicing regulatory protein (Esrp1) develop cleft lip and palate. Loss of both Esrp1 and its paralog Esrp2 results in widespread developmental defects with broad implications to human disease. Deletion of the Esrps in the epidermis revealed their requirement for establishing a proper skin barrier, a primary function of epithelial cells comprising the epidermis. We profiled the global Esrp-mediated splicing regulatory program in epidermis, which revealed large-scale programs of epithelial cell-type-specific splicing required for epithelial cell functions. These mice represent a valuable model for evaluating the essential role for AS in development and function of epithelial cells, which play essential roles in tissue homeostasis in numerous organs, and provide a genetic tool to evaluate important functional properties of epithelial-specific splice variants in vivo.

**\*For correspondence:** yxing@ucla.edu (YX); russcars@upenn.edu (RPC)

[†]These authors contributed equally to this work

**Present address:** [‡]Computer Engineering and Computer Science, Kentucky Biomedical Research Infrastructure Network, University of Louisville, Louisville, United States

**Competing interests:** The authors declare that no competing interests exist.

## Introduction

Alternative splicing (AS), the differential incorporation of exons into a mature mRNA transcript, is a widespread mechanism of generating multiple mRNAs from a single gene (*Chen and Manley, 2009*; *Kalsotra and Cooper, 2011*; *Fu and Ares, 2014*). The result of AS is the proteomic expansion of an otherwise limited genome. Switches in mRNA isoforms due to AS often result in protein isoforms with distinct functions essential for normal development, and dysregulated splicing of these events can lead to disease (*Moroy and Heyd, 2007*; *Singh and Cooper, 2012*; *Kelemen et al., 2013*; *Cieply and Carstens, 2015*). Indeed, over 95% of multi-exon genes in humans are alternatively spliced and AS has been shown to have essential functions across vertebrate and invertebrate lineages (*Graveley, 2001*; *Pan et al., 2008*; *Wang et al., 2008*). AS is regulated by a balance of positive and negatively acting RNA binding proteins (RBPs) that can alter the recruitment of the spliceosome, the macromolecular RNA-protein complex required for splicing, to the 5′ and 3′ splice sites in a pre-mRNA. Regulatory splicing factors are accessory RBPs which are often not components of the spliceosome, but that associate with a subset of transcripts via interactions with cognate RNA binding sites within exons and introns of the pre-mRNA. In many cases the position of the regulatory splicing factors relative to one another and to the alternatively spliced exon determine whether they promote splicing or skipping of a given exon (*Fu and Ares, 2014*). In addition to the accessory RNA splicing factors, core spliceosomal

**eLife digest** Genes are turned into their protein products via two steps. The first, transcription, produces an intermediate RNA molecule or 'transcript'; the second step, translation, turns the transcript into a protein. Most genes in mammals contain stretches of DNA called exons, which code for protein, interspersed with sequences called introns that do not. Therefore, a transcript must be 'spliced' before translation—the introns are removed and the exons joined.

In some genes, certain exons can be optionally included or excluded from a transcript to produce different versions of the same protein that can often have very different functions. This is known as alternative splicing, and is essential for normal development. A large number of regulatory proteins control this process, many of which are only made in specific types of cells or tissues.

Esrp1 and Esrp2 are two proteins that regulate alternative splicing in epithelial cells. These specialized cells form sheets that line most organs in the body and are found in the epidermis, the outermost layer of the skin. Although Esrp1 and Esrp2 have previously been studied in the laboratory using cultured cell lines, their roles have not been investigated in living animals.

Bebee, Park et al. have now examined mice that are unable to produce one or both of these proteins. Mice that only lacked Esrp1 developed a cleft lip and palate. In mice that lacked both proteins, many organs failed to develop correctly and in some cases did not form at all. In the epidermis, the loss of Esrp1 and Esrp2 disrupted the splicing of the transcripts from genes that give epithelial cells many of their specialized characteristics, such as the ability to form sheets of cells with well formed junctions between them. This meant that epidermis that lacked Esrp1 and Esrp2 could not form a proper barrier layer, which is a crucial role of epithelia in skin as well as in other organs.

In future, the mutant mice will be valuable for exploring how alternative splicing affects the development of epithelial cells and their properties.

proteins can also modulate the efficiency of exon recognition and in turn modify AS (*Park et al., 2004*; *Saltzman et al., 2011*).

There are several classes of regulatory splicing factors including the well-described serine/arginine rich (SR) proteins and heterogeneous nuclear ribonucloproteins (hnRNPs), which are largely ubiquitously expressed (*Fu and Ares, 2014*). However, a growing number of cell- and tissue-type-specific splicing factors have been identified, which regulate global splicing regulatory networks (SRNs) that converge on functionally coherent biological pathways required for the cell or tissue in which they are expressed (*Ule et al., 2005*; *Kalsotra et al., 2008*; *Warzecha et al., 2009a*; *Gehman et al., 2011*; *Licatalosi et al., 2012*; *Quesnel-Vallieres et al., 2015*). Transcripts that are alternatively spliced as part of tissue- or cell-type specific SRNs do not significantly overlap with transcriptional gene expression programs in the same tissues or cells, indicating that AS and transcriptional regulation are largely independent processes utilized to diversify the transcriptomes that distinguish different cell types (*Pan et al., 2008*). This observation thus highlights the function of AS as an added layer of gene expression regulation, as well as the potential role of AS in fine tuning the functions of proteins in biological pathways relevant to different cell types. Tissue- or cell-type-specific regulatory splicing factors and the programs of AS under their control are often required during normal development, which has been most extensively characterized in brain, muscle, and cardiac tissue (*Kalsotra et al., 2008*; *Gehman et al., 2011*; *Licatalosi et al., 2012*; *Lee et al., 2013*; *Li et al., 2014*; *Yang et al., 2014*; *Quesnel-Vallieres et al., 2015*). The bulk of tissue-specific splicing factors have been identified in these organ systems, and in turn the requirement of tissue specific AS in cardiac, neural, and muscle development has been further characterized in mouse models. The loss of regulatory splicing factors in these tissues often result in defects in fetal to adult splicing transitions, tissue patterning defects, and organ dysfunction (*Licatalosi et al., 2012*; *Li et al., 2014*; *Yang et al., 2014*). Tissue and/or cell-type-specific networks of AS contribute to the posttranscriptional landscape of a tissue or cell type, generating splicing isoforms that provide specific functions in the tissue or cell in which they are generated (*Castle et al., 2008*). It is important to recognize that most human and mouse tissues and organs are comprised of numerous distinct cell types. These cell types within tissues include endothelial, epithelial, and fibroblast cells that express common splicing patterns independent of the tissue of origin (*Mallinjoud et al., 2014*). This further emphasizes the role of cell

type- or tissue-specific AS outside of the nervous system, muscle, and cardiac tissues as well as within defined cell types. As such there remain many gaps in our understanding of distinct cell type-specific programs of AS that differ both between and within different tissues.

The requirement of regulatory splicing factors in normal development underscores the requirement of AS and the protein isoforms generated therein in normal patterning and development. One example of a cell-type-specific splicing event that is essential for normal development is AS of the *Fibroblast growth factor receptor 2 (Fgfr2)* pre-mRNA, where there is distinct expression of the *Fgfr2-IIIb* or *Fgfr2-IIIc* isoform in epithelial and mesenchymal cells, respectively (*Orr-Urtreger et al., 1993*). The mutually exclusive AS of exons IIIb and IIIc alters the second half of the Immunoglobulin-like III (Ig3) domain of the receptor extracellular ligand binding domain, and in turn alters the binding specificity of the Fgfr2 receptor to its cognate Fgf ligands (*Zhang et al., 2006*). The exquisite cell-type-specific expression of the Fgfr2-IIIb and Fgfr2-IIIc isoforms underlies reciprocal communication between epithelium and neighboring mesenchyme, wherein the ability of the epithelial Fgfr2-IIIb receptor to respond to mesenchymally expressed Fgfs, and vice versa, has proven essential for the development of multiple organs and appendages during embryogenesis. Knockout (KO) of *Fgfr2-IIIb* or one of its primary ligands, Fgf10, in mouse results in developmental defects in limb, lung, craniofacial and palate formation, salivary gland, tooth, and epidermal appendage (hair) formation (*Xu et al., 1998*; *De Moerlooze et al., 2000*; *Ohuchi et al., 2000*; *Revest et al., 2001*; *Petiot et al., 2003*). This highlights the crucial function of this signaling axis in normal patterning during development in a wide range of cells and tissues. Moreover, the AS of *Fgfr2-IIIb/c* has also been shown to dynamically change during the cellular transformation process of epithelial to mesenchymal transition (EMT), a necessary developmental process for embryo and tissue morphogenesis (*Savagner et al., 1994*; *Thiery et al., 2009*).

Given the distinct cell type-specific splicing of *Fgfr2* transcripts and well outlined functional consequences we and others sought to define the cis- and trans-acting determinants of this AS event (*Carstens et al., 2000*; *Hovhannisyan and Carstens, 2005*; *Mauger et al., 2008*). Using a genome-wide high-throughput cDNA screen we identified the epithelial splicing regulatory proteins 1 and 2 (Esrp1 and Esrp2) as essential regulators of *Fgfr2* splicing (*Warzecha et al., 2009a*). The expression of the ESRPs (ESRP1 and ESRP2) is epithelial cell type-specific and knockdown of ESRP1 and ESRP2 in epithelial cells induces a complete switch from *Fgfr2-IIIb* to *Fgfr2-IIIc* isoforms. Conversely, ectopic expression of Esrp1 in mesenchymal cells induces a switch from *Fgfr2-IIIc* to *Fgfr2-IIIb*, indicating that the ESRPs are necessary and sufficient to enforce expression of the epithelial *Fgfr2-IIIb* isoform (*Warzecha et al., 2009a*). Using splicing sensitive microarrays and high throughput sequencing (RNA-seq), we previously used cell lines to demonstrate that the ESRPs control a genome-wide program of AS (*Warzecha et al., 2009b*, *2010*; *Dittmar et al., 2012*). Importantly, ESRP targets are enriched for genes involved in epithelial cell properties such as cytoskeletal dynamics, cell motility, cell–cell junctions, and pathways involved in EMT. These observations strongly suggested that genome-wide programs of ESRP regulated splicing coordinate biological functions necessary for normal development and epithelial cell function (*Warzecha et al., 2009b*, *2010*; *Dittmar et al., 2012*).

While we previously used cell culture based systems to define the targets and functions of the ESRPs in vitro, a definitive investigation of their functions during development and identification of targets in vivo was needed. The ESRP mouse homologues, *Esrp1* and *Esrp2*, are specifically expressed in epithelial cells in numerous organs in adult mice and are essentially absent in mesenchymal cells and other non-epithelial cell types (*Figure 1—figure supplement 1*). For example, our previous RNA in situ RNA analysis for *Esrp1* and similar in situ analysis for *Esrp2* presented here document expression of both mRNAs in all epithelial cell layers that comprise the epidermis as well as epithelial cells of the lung, large and small intestine, salivary glands, bladder, kidney, and other prototypical epithelial cell layers (*Figure 1—figure supplement 1B–D*, also see Figures S5–S8 in *Warzecha et al. (2009a)*). Furthermore, in situ analysis of *Esrp1* expression during mouse embryogenesis demonstrated specific *Esrp1* expression in definitive endoderm and surface ectoderm as well as epithelial cells derived from these germ layers (*Revil and Jerome-Majewska, 2013*). Comprehensive microarray analysis in mouse tissues and cells has also documented the expression of both *Esrp1* and *Esrp2* in distinct tissues and cell types (*Figure 1—figure supplement 1A*) (*Su et al., 2002*). To address the functional requirement of the Esrps in development we generated *Esrp1* and *Esrp2* KO mice. Whereas *Esrp2* null mice are viable and fertile, we demonstrate that KO of *Esrp1* alone is neonatal lethal at Postnatal Day (P) 0, associated with fully penetrant bilateral cleft lip and palate (CL/P). *Esrp1* KO combined with loss of

*Esrp2* resulted in more pronounced defects in organogenesis that will be described herein. For the first in vivo evaluation of a global Esrp-regulated AS program we utilized total epidermis for RNA-seq. The loss of Esrps in epidermis is also associated with global changes in gene expression indicative of an epithelial barrier defect. We confirmed this skin barrier defect using conditional KO of the *Esrps* in epidermis. The *Esrp* KO mice reveal the essential role of the Esrps in maintaining an epithelial cell-specific splicing program for normal development during mouse embryogenesis, underscoring the role of AS in the development of a multitude of organs and appendages, and the requirement of the Esrps in critical epithelial cell functions.

## Results

### Gross morphological defects in Esrp KO mice

To characterize the functions of Esrp1 and Esrp2 in vivo during mammalian development we generated mice with conditional and/or germline *Esrp* KO alleles. Our previous cell culture based systems indicated that Esrp1 was the primary driver of ESRP-regulated splicing, and thus we hypothesized Esrp1 would be required during development. We generated a conditional *Esrp1* KO allele wherein exons 7–9 were floxed. Mice with germline *Esrp1* KO were generating in crosses with mice carrying *Zp3-Cre* transgenes. Excision of exons 7–9 following Cre-mediated recombination removes the first RNA Recognition Motif (RRM), and places the remaining transcript out of frame, generating a null allele of *Esrp1* (*Figure 1—figure supplement 2A–D*). Since we did not anticipate that *Esrp2* KO alone would yield a phenotype we obtained an *Esrp2* KO mouse ES cell line from the Knockout Mouse Project (KOMP) (*Figure 1—figure supplement 2E*).

To evaluate phenotypes associated with germline ablation of the Esrps we crossed mice carrying KO alleles for *Esrp1* and *Esrp2* to generate *Esrp1* KO (*Esrp1$^{-/-}$*, *Esrp2$^{+/+}$*) (KO), *Esrp1* KO *Esrp2* heterozygous (*Esrp1$^{-/-}$*, *Esrp2$^{+/-}$*) (KH), and *Esrp1* and *Esrp2* double KO (*Esrp1$^{-/-}$*, *Esrp2$^{-/-}$*) (DKO) mice. From the crosses we generated nine allelic combinations, including all combinations of *Esrp1* and *Esrp2* WT and KO alleles. We confirmed the expected Mendelian transmission rates of the *Esrp1* and *Esrp2* KO alleles in these crosses, indicating there was no early embryonic lethality associated with loss of the Esrps. As we predicted, *Esrp2* null mice were viable and fertile without any detectable abnormalities. In contrast, mice with germline KO of *Esrp1* exhibited fully penetrant bilateral cleft lip and cleft palate (CL/P), and these mice are neonatal lethal on post natal day 0 (P0) most likely due to orofacial clefting. Therefore, while Esrp2 is not essential in mice with intact *Esrp1* alleles, Esrp1 is necessary for craniofacial development and post-natal viability.

Because *Esrp1* ablation was neonatal lethal, further analysis of developmental defects in Esrp deficient mice was carried out in embryos. We harvested E18.5 embryos from all nine genetic combinations. Overall embryo metrics including weight and crown to rump length showed KO embryos were normal in size, KH embryos were only slightly reduced in weight and length compared to WT littermate embryos, whereas DKO embryos showed a ~30% reduction in length (*Figure 1—figure supplement 3D,E*). To characterize the CL/P phenotype we evaluated the gross morphology of post-fixed E18.5 embryo heads. Esrp1 KO embryos exhibited bilateral cleft lip and clefting of the primary and secondary hard palate, as well as the soft palate (n > 40 embryos evaluated). This phenotype was also observed in KH and DKO embryos, but these embryos also exhibited more pronounced craniofacial defects including profound rostral shortening, and mandibular hypoplasia and dysplasia (*Figure 1A–L*). We performed Alizarin Red and Alcian Blue bone and cartilage stains to evaluate the skeletal defects associated with the CL/P phenotype. These stains further defined the rostral shortening defect and mandibular defects in the Esrp deficient embryos. DKO mice presented with the most severe defects, resulting in a malformed mandible lacking the normal curvature seen in control embryos (*Figure 1M–P,U–X*). Closer inspection of the ventral aspect of cranial bone stains revealed absence of the premaxillary bones and palatal bone hypoplasia, consistent with CL/P (*Figure 1Q–T*, *Figure 1—figure supplement 3A*, dorsal aspect). The mandibles of Esrp KO embryos lacked the coronoid process (*Figure 1U–X*), which explains the widened oral opening in the Esrp deficient mice. These results support a requirement for Esrp1 in normal craniofacial patterning and development. These studies also demonstrate that Esrp2 can partially compensate for loss of Esrp1, as craniofacial phenotypes observed in *Esrp1* KO mice are more profound in combination with deletion of one or both alleles of *Esrp2*.

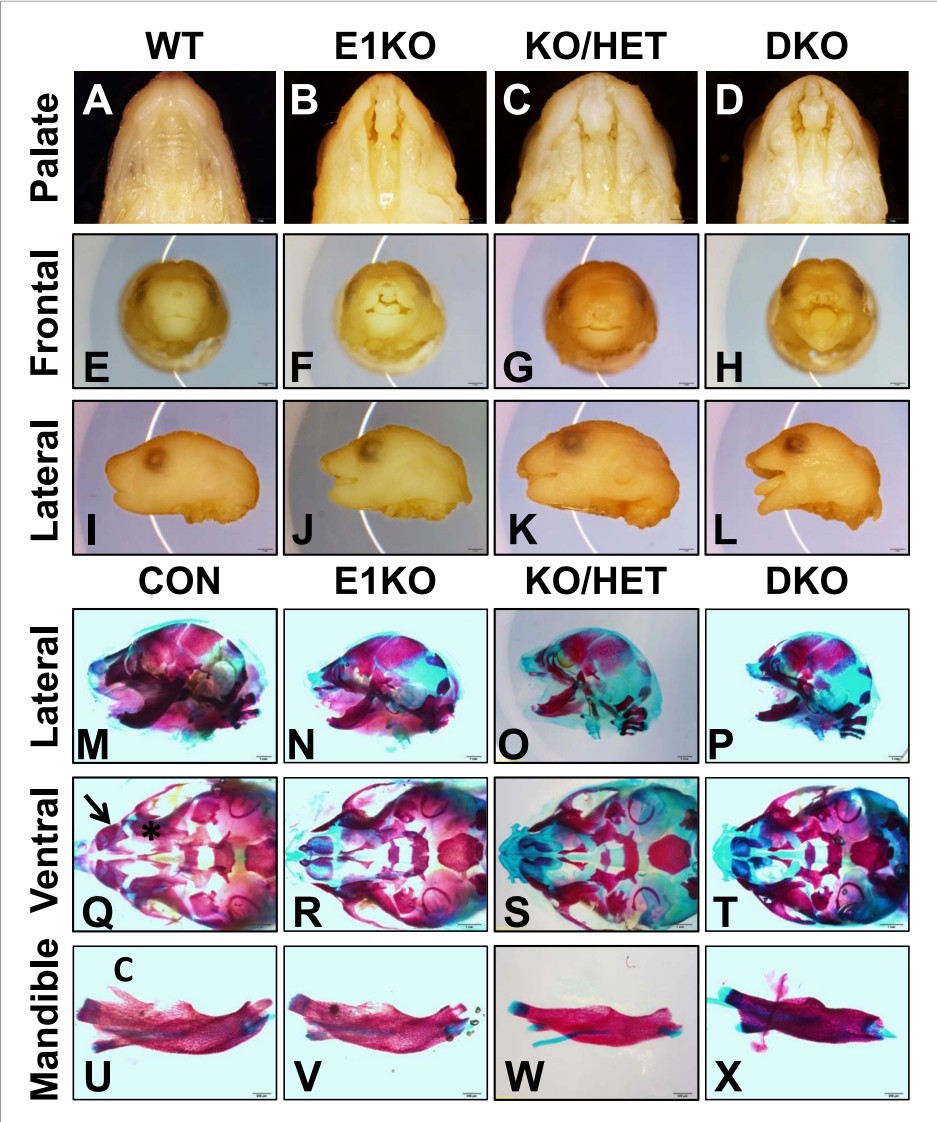

**Figure 1**. Esrp knockout (KO) mice have defects in craniofacial development. (**A–D**) E18.5 palatal images of Esrp KO mice reveal bilateral cleft lip and cleft palate (CL/P) (n = 4 for each genetic group). (**E–L**) Frontal (**E–H**) and lateral (**I–L**) images Esrp KO mice show bilateral cleft lip and rostral malformation/foreshortening (n = 4 for each genetic group). (**M–X**) Alizarin Red/Alcian Blue Bone and cartilage stains of E18.5 embryos (n = 3 for each genetic group). (**M–P**) Esrp KO mice exhibit rostral shortening and mandibular malformation. (**Q–T**) Esrp KO mice have palatal hypoplasia and absence of premaxillary bones consistent with CL/P. Premaxilla bone (arrow in **Q**) and palatal bone (* in **Q**) are indicated in control embryos. (**U–X**) Mandibles of Esrp KO mice lack the coronoid process (**C**). DKO mandibles are profoundly malformed (**X**).

The following figure supplements are available for figure 1:

**Figure supplement 1**. *Esrp1* and *Esrp2* expression in tissues and cells of embryonic and adult mice.

**Figure supplement 2**. Generation of Esrp1 and Esrp2 KO alleles.

**Figure supplement 3**. Phenotypes in Esrp KO embryos.

**Figure supplement 4**. Organogenesis defects in Esrp deficient embryos.

Additional evidence that Esrp2 can partially compensate for Esrp1 KO was observed when we analyzed limb phenotypes in our mice. Whereas we observed no gross forelimb or hindlimb defects in *Esrp1* KO mice, Esrp KH embryos often presented with malformed forelimbs and forepaw syndactyly (n = 22 evaluated). In DKO E18.5 embryos the forelimbs were further malformed, associated with smaller limbs and full fusion of the forepaw, or partially penetrant forelimb agenesis (∼33%, n = 9/27 evaluated). Bone and cartilage stains revealed the forelimb defect was associated with the absence of the radius and humerus bones (*Figure 1—figure supplement 3B,C*). The forelimb defects thus also represent an example where *Esrp2* can partially compensate for *Esrp1* ablation, and that *Esrp2* can modify the phenotype in a dose dependent manner. The variable penetrance seen in the DKO forelimbs, in which forelimb agenesis or profound limb malformation were seen, further underscores the crucial function of Esrps in limb development. While the mechanisms that give rise to these abnormalities require further study, *Esrp1* and *Esrp2* are expressed in the ectoderm overlying mesenchyme of the apical ectodermal ridge (AER) and crosstalk between these cells, including a role of Fgfr2-IIIb in ectoderm, have been shown to underlie limb development (*Xu et al., 1998*). However, hindlimbs were unaffected in Esrp KO mice and therefore the molecular cues driving formation of forelimb and hindlimbs clearly differ with respect to a requirement for Esrp regulated splicing.

## The Esrps are required for organogenesis of multiple organ systems

As noted previously, the Esrp proteins were identified as primary regulators of the *Fgfr2-IIIb* splicing event (*Warzecha et al., 2009a*). Previous work that evaluated the developmental requirement for exon IIIb in mice used a mouse allele in which exon IIIb was deleted. The removal of exon IIIb in the mature transcript results in mice with a multitude of developmental defects in a variety of organ systems during embryonic development including lung, thymus, salivary gland, and kidney (*De Moerlooze et al., 2000*). To determine if the loss of the Esrp proteins impacted development of many of these organs, we collected whole embryo sagittal sections from Esrp1 KO and DKO embryos and littermate controls at E15.5. We observed agenesis of two organs that undergo branching morphogenesis, lung and salivary gland, in the DKO embryos (*Figure 1—figure supplement 4*). The failure to form lung and salivary glands in DKO embryos represents examples of organ development that require the Esrp proteins, but where partial compensation for *Esrp1* ablation by a single copy of *Esrp2* is sufficient for these organs to form. This implicates the Esrps in regulating branching morphogenesis in vivo, but not all organs utilizing this developmental program are absent in Esrp DKO mice. Kidneys, which undergo a stereotypical branching morphogenesis, are present in the KO and DKO embryos, albeit reduced in size. This level of histological analysis cannot rule out potential subtle defects in branching morphogenesis that may impact organ size in the kidney. Further detailed analysis of these phenotypes will be needed to evaluate the developmental timing and levels of branching defects in Esrp deficient mice.

## Esrp1/2 DKO embryos exhibit epidermal hypoplasia and hair follicle developmental defects

Previous studies have shown that the epithelial cells that comprise the skin exhibit high levels of *Esrp1* and *Esrp2*, both in the interfollicular epidermis as well as the epithelial cells of the hair follicle (*Figure 1—figure supplement 1A,E*) (*Su et al., 2002*; *Rendl et al., 2005*; *Zhang et al., 2008*; *Greco et al., 2009*; *Warzecha et al., 2009a*). The skin forms an essential barrier that protects the body from environmental stresses and hazards, maintains fluid electrolyte homeostasis, and provides a thermoregulatory layer. We therefore focused further histologic and molecular analysis on KO and DKO skin. We harvested E18.5 dorsal skin and performed H&E staining of dorsal midline sagittal sections. Skin isolated from KO and KH embryos did not reveal any major defects in skin formation or appendages (i.e.,: hair follicles). However, DKO embryos had thin transparent skin at E18.5. Histological analysis of DKO and control littermates revealed epidermal hypoplasia (*Figure 2A*). Quantification of epidermal thickness, as measured from the basal keratinocyte layer to the granular layer, in DKO skin was reduced by 35% compared to littermate controls carrying at least one wild type copy of *Esrp1* (Control: 25.79 ± 2.20 μm, n = 8, DKO: 16.68 ± 1.66 μm, n = 6, unpaired two-tailed t test p < 0.0001) (*Figure 2B*). In addition to epidermal hypoplasia, DKO skin at the histological level appeared to have reduced hair follicle number. To address this we counted hair follicle numbers over 9 fields of view along the length of the skin. In DKO skin there were ∼23% fewer hair follicles (Control:

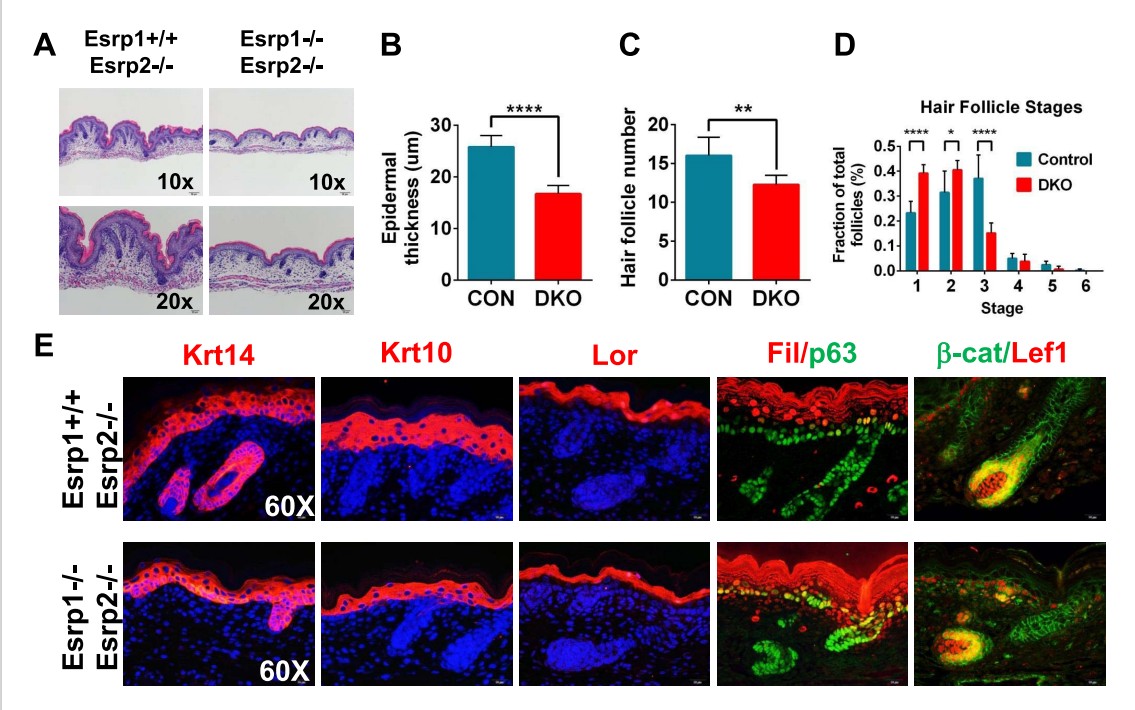

**Figure 2**. KO of Esrp1 and Esrp2 results in defects in epidermal and follicular development. (**A**) Representative H&E stained sections of dorsal skin from control *Esrp1+/+, Esrp2−/−* and *Esrp* DKO (*Esrp1−/−, Esrp2−/−*) E18.5 embryos used in **B**–**E**. (**B**–**D**) Metrics of epidermal thickness measured from basal keratinocyte layer to granular layer (**B**), hair follicle number measured over 9 fields of view (**C**), and hair follicle stages from Esrp DKO (n = 6) and control (CON) (n = 8) littermates. Two-tailed Student's *t*-test was used for **B**, **C** and 2way ANOVA multiple comparisons test for **D**. (**E**) Immunofluorescence of skin differentiation markers for basal keratinocytes (Krt14 and p63), spinous layer (K10), cornified layer (Loricrin (Lor)), and granular layer (Filaggrin (Fil)). β-catenin and its transcriptional target Lef1 are markers of developing hair follicles (n = 3).

The following figure supplement is available for figure 2:

**Figure supplement 1**. KO of Esrp1 and Esrp2 results in defects in epidermal development and hair follicle formation.

16.00 ± 2.37, n = 8, DKO: 12.25 ± 1.24, n = 6, unpaired two-tailed t test p = 0.0043) (*Figure 2C*). Moreover, the reduced number of hair follicles in DKO skin was associated with earlier staged hair follicles compared to controls, indicating delayed hair follicle maturation (*Figure 2D*) (*Paus et al., 1999*; *Petiot et al., 2003*).

While the skin appeared to contain all stratified layers of the epidermis, it was possible that the observed epidermal hypoplasia might be due to under representation of one or more of these layers. Mouse epidermis is comprised of interfollicular epidermis, hair follicles, and sebaceous glands. Basal keratinocytes reside in the deepest layer of the epidermis adjacent to the underlying dermis and are arranged in a single cell layer, both within the interfollicular epidermis and hair follicles. The source of stem cells needed for vertical proliferation and differentiation for stratification of the epidermis reside within the basal keratinocyte layer. These stem cells give rise to transit amplifying cells that vertically proliferate and differentiate into the other stratified epidermal layers (spinous layer, granular layer, and stratum corneum) (*Fuchs, 2007*; *Solanas and Benitah, 2013*). To evaluate the differentiation program in our *Esrp* DKO mice we performed indirect fluorescent immunohistochemistry for epidermal differentiation markers. We used markers of basal keratinocytes (Keratin 14, p63), spinous layer (Keratin 10), granular layer (Loricrin, Filaggrin), and hair follicles (β-catenin, Lef-1) in E18.5 skin sections (*Figure 2E*). DKO skin sections stained positively for these differentiation markers; however, Keratin 14, Keratin 10, Loricrin, and Filaggrin expression domains were reduced in overall size compared to the control skin samples (*Figure 2E* and *Figure 2—figure supplement 1*). This is consistent with the observed epidermal hypoplasia, and illustrates that DKO epidermis nonetheless is still capable of forming a stratified epidermis. Where hair follicles were present, we observed cortical

cytoplasmic β-catenin in the epithelial cells of the hair follicle. Nuclear expression of β-catenin was observed in cells bordering the dermal cells, which expressed nuclear Lef-1, a β-catenin transcriptional target. Thus, hair formation was also functionally intact, although overall follicle number was reduced in DKO skin. We also noted that p63 staining reveals a less organized basal keratinocyte layer, with disorganized and sporadically spaced p63 negative cells (*Figure 2E*, *Figure 2—figure supplement 1*). These skin phenotypes are similar to those previously observed in Fgfr2-IIIb isoform-specific KO mice that also showed epidermal hypoplasia and reduced and immature hair follicle numbers (*Petiot et al., 2003*).

## Esrp regulated splicing targets exhibit variable sensitivity to loss of Esrp1 and Esrp2

The progression of phenotypes observed in combined *Esrp1* and *Esrp2* DKO mice compared to *Esrp1* KO alone implies some functional redundancy of these two splicing factors. In turn, this suggests the degree to which Esrp2 can partially compensate for Esrp1 KO contributes to the observed defects in the mice. To address this possibility we utilized the most readily abundant source of epithelial cells that can be purified to relative homogeneity, the epidermis, to define differential changes in splicing associated with loss of *Esrp1* and/or *Esrp2* alleles. We isolated full thickness epidermis from E18.5 embryonic mice following trypsin digestion and manual separation from the underlying dermis. Total epidermis from the nine genetic combinations of *Esrp1* and *Esrp2* KO alleles were isolated to evaluate *Esrp1* and *Esrp2* expression levels. Quantitative real-time PCR (*Figure 3A*, graph) and western blot analysis for Esrp1 and Esrp2 confirmed transcript and protein ablation in vivo (*Figure 3A*, bottom panel).

To address differential responses to loss of Esrp1 and/or Esrp2, we performed semi-quantitative radioactive RT-PCR for the mutually exclusive splicing of *Fgfr2-IIIb/c*. Across all nine allelic combinations no change in *Fgfr2-IIIb* (epithelial) splicing was detected until both Esrp1 and Esrp2 were deleted, and thus only a single copy of Esrp2 is sufficient to maintain *Fgfr2-IIIb* splicing in epidermis. However, in DKO epidermis *Fgfr2* underwent a full switch from *Fgfr2-IIIb* (epithelial) to *Fgfr2-IIIc* (mesenchymal), indicating that the production of the Fgfr2-IIIb isoform is fully dependent upon the Esrps in epidermis (*Figure 3B*). We tested whether the Esrps also regulate *Fgfr1* and *Fgfr3*; transcripts that also contain mutually exclusive exons (MXEs) IIIb and IIIc in the same protein coding region. In both *Fgfr1* and *Fgfr3*, we also observed a complete switch from exon IIIb to exon IIIc splicing in DKO epidermis. However, *Fgfr1* showed a more dramatic switch in splicing in *Esrp1* KO epidermis than *Fgfr3*, but both showed a gradual switch to the *-IIIc* isoforms upon sequential loss of the Esrps (*Figure 3B*). This places the Esrps central in the regulation of this developmentally significant family of tyrosine kinase receptors, where loss of both Esrp1 and Esrp2 results in a splicing switch to the mesenchymal receptor isoforms in the epidermis.

To determine if other Esrp splicing targets are differentially affected by loss of Esrp1 and Esrp2, in a manner similar to the *Fgfr* family, we tested splicing events in mouse orthologs of previously described splicing events from our in vitro human cell culture experiments (*Warzecha et al., 2009a*, *2009b*). We evaluated an Esrp enhanced splicing event in *Enah* and the Esrp silenced splicing event in *Arhgef11*. *Enah* is more profoundly affected by loss of Esrp1, while *Arhgef11* remains relatively unchanged until the KH and DKO samples (*Figure 3C*). These results indicate differential sensitivity of Esrp regulated splicing events to loss of Esrp1 and Esrp2, and thus variable functional redundancy of Esrp1 and Esrp2 in regulation of Esrp-mediated splicing events.

## In vivo determination of a global Esrp regulated splicing program in the epidermis

To extend our understanding of ESRP regulated splicing events beyond those previously identified in vitro and to determine the splicing regulatory program mediated by the Esrps in vivo, we profiled global splicing changes using RNA isolated from total epidermis from WT, KO, KH, and DKO E18.5 embryos. From these four genetic groups we sought to identify global changes in AS that show variable sensitivity to loss of Esrp1 and Esrp2, as was seen in *Fgfr1*, *Fgfr2*, *Fgfr3*, *Enah*, and *Arhgef11*. Each library was constructed using total RNA from full thickness epidermis to generate strand-specific polyA-selected mRNA sequencing libraries and the multiplexed libraries were sequenced together on a single flow cell across 2 lanes with an Illumina HiSeq 2000 (see also 'Materials and

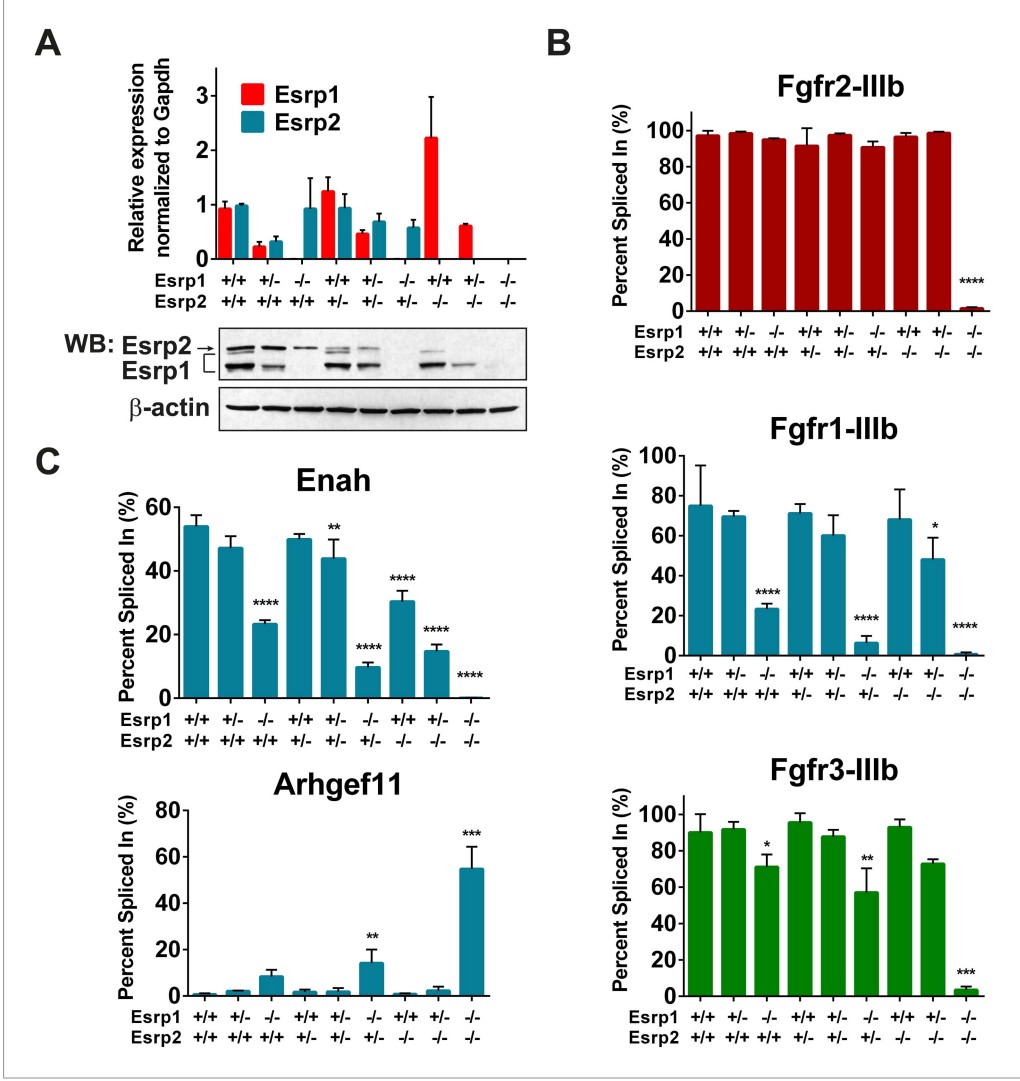

**Figure 3**. Esrp regulated splicing events show variable sensitivity to loss of Esrp1 and Esrp2. (**A**) qRT-PCR expression of *Esrp1* and *Esrp2* in purified E18.5 epidermis from the designated genetics of Esrp1 and Esrp2 KO embryos (n = 3). Western blot confirmation of Esrp1 and Esrp2 KO in purified epidermis from E18.5 embryos (n = 2). (**B**) Graphical representation of the epithelial -IIIb exon inclusion rates for *Fgfr1*, *Fgfr2*, and *Fgfr3* in epidermis (n = 3). (**C**) Esrp regulated splicing events in Enah and Arhgef11. Graphical representation of Percent Spliced in (PSI) are presented (n = 3). Two-way ANOVA multiple comparisons tests statistical analysis was used and all groups were compared to $Esrp1^{+/+}$, $Esrp2^{+/+}$ (WT). Statistical indications for p-values, *<0.05, **<0.01, ***<0.001, ****<0.0001.

methods'). We generated over 560 million paired sequencing reads in total from 2–3 biological replicates from each genetic group which uniquely mapped to over 440 million mapped pairs. Each genetic group was represented by at least 70 million unique paired sequencing reads mapped to the mouse mm10 genome at ~70–85% mapping rate (*Supplementary file 1*). The RNA sequencing results were subsequently processed by rMATS (*Shen et al., 2014*) to identify differential splicing events.

Each KO group was compared to the WT group to identify differentially spliced events with an associated change in Percent Spliced In (PSI, ΔPSI or Δψ) of these events. The detected AS events from the four genetic groups were clustered for each comparison and a heat map was used to represent ΔPSI in the three comparisons of the different KO conditions relative to WT controls. As expected, the largest number and greatest ΔPSI in detected splicing switches were seen in the DKO epidermis (*Figure 4A*). The splicing events for the five major classes of AS were cataloged: Skipped

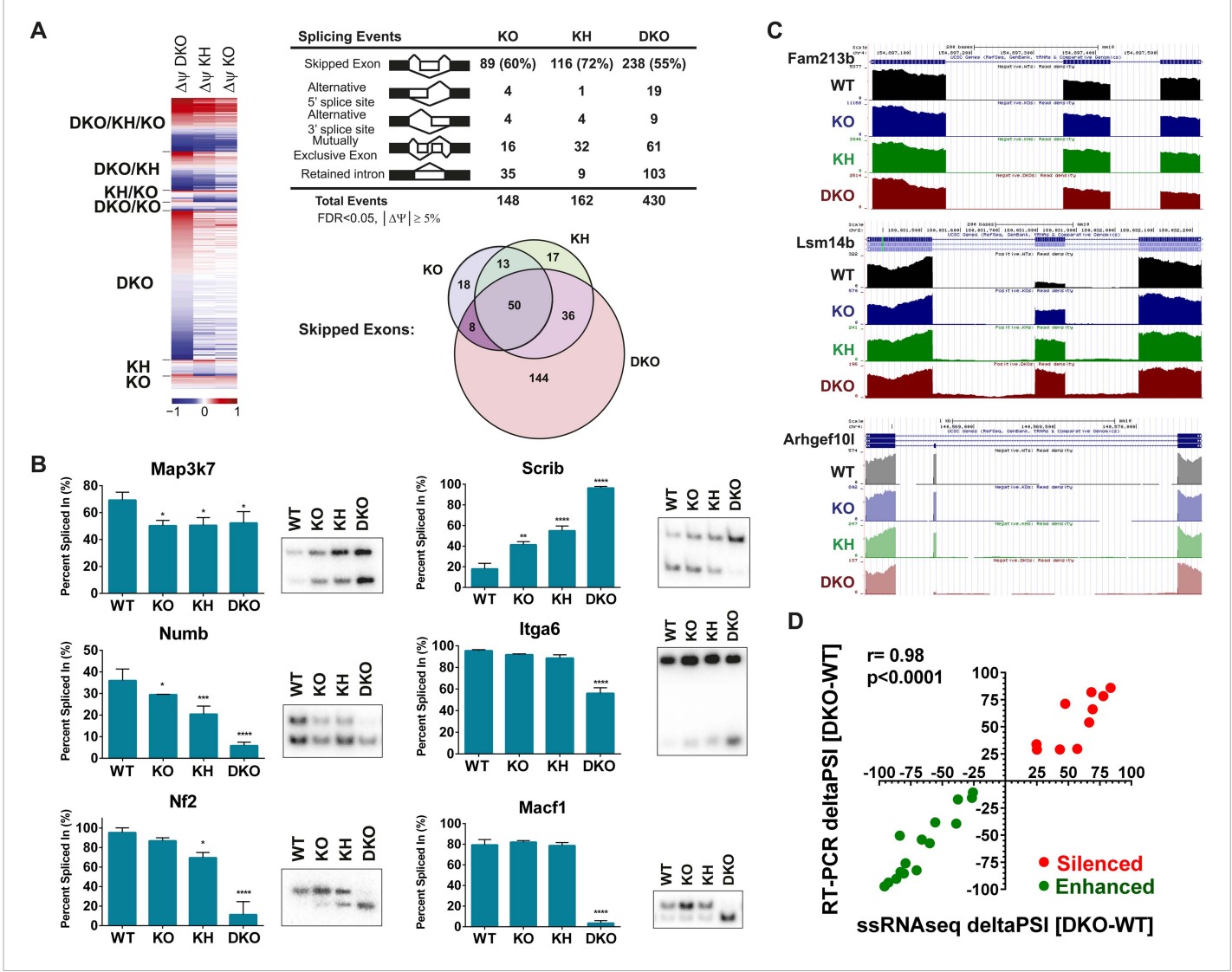

**Figure 4.** Global analysis of Esrp-mediated alternative splicing (AS) in epidermis. (**A**) Heatmap representing the predicted ΔPSI values for skipped exon (SE) events from KO, KH, and DKO epidermis compared to WT. Summary table of total detected splicing events in KO, KH, and DKO strand-specific RNA-seq. Venn diagram depicting overlap in detected SE events in KO, KH, and DKO datasets. Splicing events detected by rMATS at a FDR < 5% and |deltaPSI| ≥ 5% are depicted here. (**B**) Graphs for PSI for six Esrp targets are shown. Representative radioactive RT-PCR PAGE gels are presented. All events measured are from three independent biological samples. Two-way ANOVA multiple comparisons tests statistical analysis was used and all groups were compared to Esrp1[+/+], Esrp2[+/+] (WT). (**C**) Representative UCSC custom genome browser snapshots of Esrp1 KO full switch (*Fam213b*), gradual (*Lsm14b*), and DKO only (*Arhgef10l*) SE splicing events. Negative strand transcripts are shown in faded colors compared to bold colored positive strand transcripts. (**D**) Graphical representation of the predicted deltaPSI of 25 ssRNA-seq targets from WT vs DKO rMATS analysis, compared to the RT-PCR validated deltaPSI. Pierson Correlation with r- and p-values is indicated.

The following source data and figure supplements are available for figure 4:

**Source data 1**. rMATS analysis of Esrp deficient epidermis.

**Source data 2**. RNAseq and RT-PCR validated SE splicing events.

**Figure supplement 1**. The Esrps regulate AS of the *Fgfr* family and *Cd44* genes in epidermis.

**Figure supplement 2**. RT-PCR validation of Esrp regulated SE splicing events.

Exon (SE, cassette exons), Alternative 5′ splice site (A5SS), Alternative 3′ splice site (A3SS), MXE, and Retained Introns (*Figure 4A*). The majority of AS events were identified in the SE category and comprise 55–72% of the total detected Esrp-dependent AS (*Figure 4A*). The Venn diagram depicting the overlap between the KO, KH, and DKO groups further highlights the growing number of SE events detected when one or both alleles of *Esrp2* are ablated in addition to *Esrp1*. We did note 48 events detected in the KO and/or KH analysis that were not identified by rMATS in the DKO samples. We suspect that some of these arise due to statistical limitations in RNA-seq based detection of AS events in addition to the higher numbers of mapped reads from the KO samples. To evaluate this further, we also noted that 14 of these 48 events arose from the 10 tandem variable exons of Cd44. The variable inclusion of these exons in the KO and KH epidermis compared to the almost full skipping of all 10 exons (Cd44s) in the DKO is a primary source of the detected non-overlapping SE events (*Figure 4—figure supplement 1*). It is also possible that some of these events are false positives or that differences in RNA sample quality or that of the libraries account for the lack of detection of some of these events in DKO samples. While the SE category represents the largest group of splicing events, there were several key splicing events identified in the MXE category as well as complex splicing patterns harder to categorize in a single group. For example, *Fgfr1*, *Fgfr2*, and *Fgfr3* were found within the top statistically significant events for the MXE category from the WT vs DKO comparison. Custom genome browser views of these three MXE events confirm the splicing patterns as determined by RT-PCR for the *Fgfr(s)* (*Figure 3B*, *Figure 4—figure supplement 1*). *Cd44* AS, mentioned previously, represents a complex conserved splicing event in which inclusion of 10 exons in the mouse variable region of the *Cd44* transcript is associated with the epithelial isoforms (*Cd44v*), and the exclusion of these exons generates the mesenchymal *Cd44s* isoform (*Brown et al., 2011*). The Esrps were shown to be primary regulators of the CD44v isoform, where ESRP depletion led to a nearly complete switch towards the mesenchymal CD44s isoform in human cell lines (*Warzecha et al., 2009a*). AS of many of these variable exons were detected in the SE category and were confirmed in the custom genome browser and by RT-PCR (*Figure 4—figure supplement 1*). We extended our analysis of variable splicing sensitivity to six additional previously known Esrp targets in WT, KO, KH, and DKO epidermis. From this expanded list of Esrp targets we observed three classes of splicing events: a full change in splicing with *Esrp1* KO with no additional splicing change in KH or DKO samples (*Map3k7*), a gradual splicing switch seen from *Esrp1* KO to KH to DKO (*Numb*, *Nf2*, and *Scrib*), or changes in splicing only observed in the DKO (*Itga6* and *Macf1*) (*Figure 4B*).

We used RT-PCR to evaluate additional Esrp regulated splicing targets for differential sensitivity to Esrp1 and Esrp2 ablation, including novel in vivo Esrp targets from the rMATS global analysis of AS in the epidermis. The splicing events selected had a minimal predicted ΔPSI > 10%, but included a broad range of PSI(s). In total we have tested 38 splicing events including the events described above (*Figure 4—source data 2*). Of the targets evaluated with successful RT-PCRs all exhibited the expected splicing pattern as predicted by rMATS (*Figure 4—figure supplement 2*). These identified targets provide further evidence of differential splicing sensitivity to loss of Esrp1 and Esrp2. All three patterns of differential sensitivity to *Esrp1* and *Esrp2* KO were seen as previously described, and the PSI values of these validations are graphically represented (E1KO, gradual, and DKO sensitive splicing changes organized from left to right) (*Figure 4—figure supplement 2A,B*). Representative RNA-seq genome browser views of Esrp regulated splicing events that undergo full switch after *Esrp1* KO (*Fam213b*), gradual splicing switch (*Lsm14b*), or no switch until DKO (*Arhgef10l*) further illustrate these patterns of Esrp regulated AS confirm these findings via RNA-seq (*Figure 4C*). Of note, we also tested 3 transcripts (*Exoc1*, *Golga2*, and *Stx2*) that were identified by rMATs in KO or KH samples, but not in DKO and noted that all three were validated to switch splicing in DKO samples (*Figure 4—figure supplement 2C*). These examples thus further indicate that there are some false negatives in the RNA-Seq analysis of DKO epidermis. Comparison of the predicted ΔPSI as measured by rMATS and the ΔPSI determined by semi quantitative RT-PCR revealed a high correlation between these two measurements in WT and DKO samples (r-value = 0.98, p < 0.0001, Pearson Correlation) (*Figure 4D*). *Grhl1 was* amongst the target SE events that showed no detectible change in PSI until DKO. The *Grhl1* SE event is also an example of predicted AS-mediated non-sense mediated decay (AS-NMD) (*Lu et al., 2015*). The inclusion of exon 5 maintains *Grhl1* expression, whereas skipping as seen in the DKO epidermis results in a transcript targeted by NMD. In DKO epidermis there is an 85% reduction in exon 5 inclusion that results in an 86% reduction in *Grhl1* transcript levels (*Figure 4—figure supplement 2A,B*, see also Figure 6C).

To determine the biological processes and pathways targeted by the Esrp regulated splicing program in vivo, we used all Esrp regulated SE events from the three WT vs Esrp KO comparisons to perform gene ontology (GO) and pathway analysis using DAVID 6.7 (*Huang da et al., 2009*) (*Supplementary file 2*). Esrp regulated splicing targets were enriched for the functional GO terms 'cytoskeletal protein binding', 'cell cortex', 'vesicle-mediated transport', and 'actin binding' as well as the KEGG pathway terms 'adhererns junction' and 'MAPK signaling pathway' (FDR [Benjamini-Hochberg] <0.05). However, several interesting functional GO terms, with p-values <0.01 while not reaching the threshold for multiple comparisons, in biological functions associated with tissue morphogenesis, cellular adhesion, membrane organization, and apoptosis. Our KEGG pathway analysis also enriched for the terms 'tight junction (TJ)', 'regulation of actin cytoskeleton', 'hematopoietic cell lineage', and 'gap junction' with p-values <0.01 while not reaching the threshold for multiple comparisons. The enrichment of these biological processes and pathways support an important functional role of the Esrps in regulated splicing of genes in pathways necessary for proper epithelial cell maintenance and function.

## Esrp binding motifs are enriched flanking Esrp regulated exons in mouse epidermis

RBPs that regulate AS are believed to predominantly act through direct binding to regulatory RNA sequence motifs in the exons and/or flanking intronic sequences of alternatively spliced exons (*Fu and Ares, 2014*). However, splicing factors may also indirectly induce changes in splicing patterns by a number of mechanisms, including changes in splicing that alter the expression level, activity, or localization of other splicing factors. To investigate the possibility that differential expression of other splicing factors in Esrp depleted epidermis might be involved in the observed regulation, we focused on a set of 188 mouse orthologues of human RBPs as well as other genes with known or putative roles in splicing regulation (*Han et al., 2013*). Since the number and the degree of splicing change were largest in Esrp DKO epidermis we analyzed differential expression of splicing factors in WT and DKO skin. We noted large changes in *Esrp1* and *Esrp2* expression in DKO epidermis, but there were some more modest changes in expression of other candidate splicing factors with statistically significant changes in expression following Esrp ablation (*Figure 5A*, *Figure 5—source data 1*). Hence, while a loss of Esrp expression is likely to play a major role in splicing of numerous transcripts, alterations in expression of some of these other candidate regulators may also contribute to splicing shifts in some targets. It is noteworthy that although *Esrp1* RNA levels are not completely absent in the *Esrp1* KO RNA-seq samples by FPKM, mapped reads from the RNA-seq data confirmed that these RNAs represent transcripts splicing across the exon 7–9 deletion from exon 6 to exon 10, which would not encode a functional Esrp1 protein and reduced transcript levels are indicative of targeting by NMD. We confirmed the loss of *Esrp1* expression by qRT-PCR targeting the exon 7–9 deletion in Esrp KO samples by western for Esrp1 in total epidermis as well as histologically in skin sections using in situ hybridization for *Esrp1* (*Figure 3A* and *Figure 1—figure supplement 1D*). The changes in gene expression of RBPs or other splicing factors in the DKO epidermis were also evaluated in KO and KH RNA-seq (*Figure 5—figure supplement 1*).

To further investigate the degree to which the SE exons were directly regulated by the Esrps in vivo, and to identify candidate splicing factors that potentially co-regulate them, we looked at known splicing factor binding motifs as well as all 6-mer motifs within or flanking the SE splicing events. The motif enrichment of known splicing factor binding sites consisted of a set of binding sites for 115 RBPs collected from the literature (see 'Materials and methods'). For each of the motifs we looked for their enrichment in the regulated exon and 250 bp of flanking sequence in the upstream and downstream intronic sequence of SE events (ΔPSI ≥ 0.05, FDR < 0.05), excluding the splice sites, and compared the motif enrichment to a background set of non-regulated splicing events (min PSI > 0.15, max PSI < 0.85, FDR > 50%, FPKM > 5.0) from the rMATS analysis (*Figure 5B*, *Figure 5—source data 2*). For this analysis we collected all significant SE events and divided them into Esrp enhanced (high PSI in WT) or silenced (high PSI in KO) from each of the pairwise comparisons of WT vs KO, KH, or DKO epidermis. Motif enrichment analysis of known splicing factor binding sites identified the top ranked Esrp1 (UGGUGG) binding site motif defined by SELEX in the upstream intron of Esrp silenced SE events in all three pairwise comparisons, and the Esrp1 motif was also enriched in the exon body of Esrp1 KO splicing targets (*Dittmar et al., 2012*). We also observed enrichment of the Esrp1 motif in the downstream intron of Esrp enhanced exons in the KO and DKO analysis. This is consistent with the

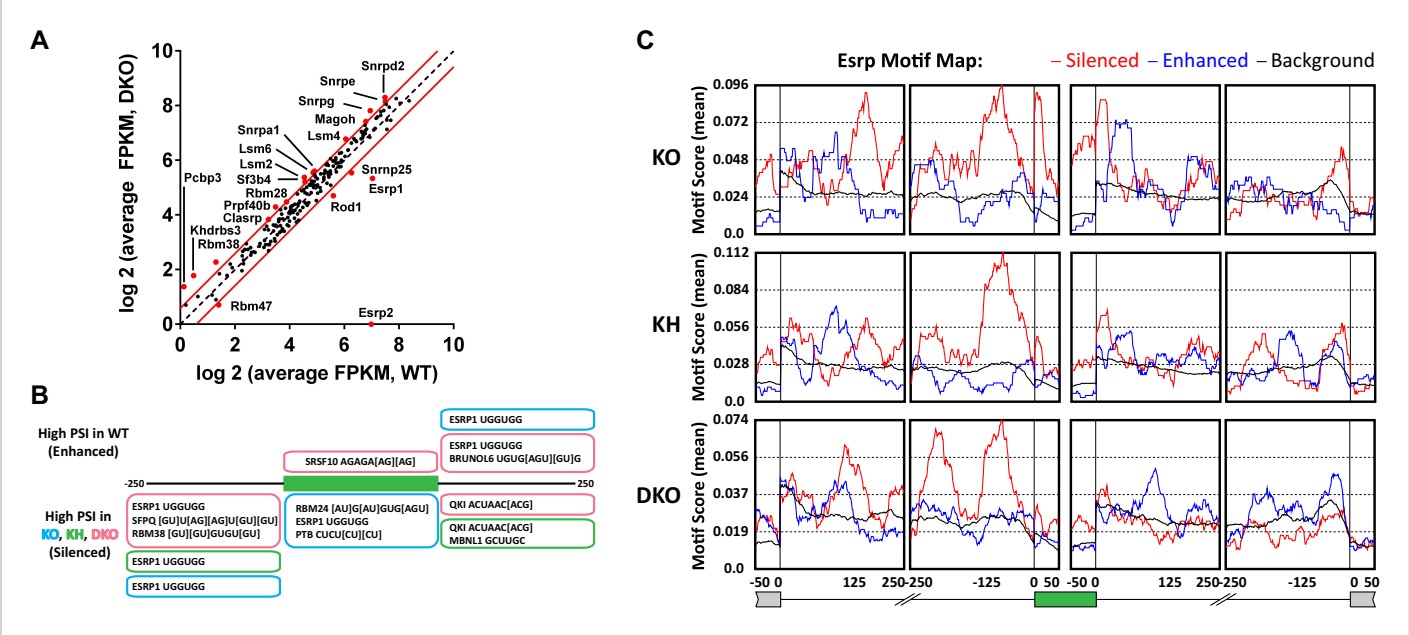

**Figure 5**. Esrp and known splicing factors RNA binding motif enrichment in Esrp-regulated SE events. (**A**) Differential gene expression of 188 mouse orthologues of known human RNA binding proteins (RBPs) or other putative splicing factors with at least 1.5-fold change in expression. Gene expression levels are from Cuffdiff v2.2.0. (**B**) Motif enrichment for RNA binding motifs of known splicing factors for SEs identified from KO (blue), KH (green), or DKO (pink) rMATS analysis. (**C**) RNA binding map for the top 12 Esrp bound 6-mer motifs identified by SELEX-seq. Maps for Esrp motif enrichment for SE events from the KO, KH, and DKO rMATS analysis are shown for silenced (red) and enhanced (blue) splicing events.

The following source data and figure supplement are available for figure 5:

**Source data 1**. Differential gene expression of splicing factors in Esrp deficient epidermis.

**Source data 2**. Enriched known RNA binding motifs in Esrp regulated SE splicing events.

**Source data 3**. Enriched 6-mers in Esrp regulated SE splicing events.

**Figure supplement 1**. Differential gene expression of RBPs in Esrp KO and KH epidermis.

Esrp RNA binding map identified from our in vitro Esrp targets, wherein Esrp binding in the upstream intron or in the exon body of a regulated SE events mediates exon skipping, and conversely binding in the downstream intron promotes exon inclusion (*Dittmar et al., 2012*). We also identified motifs for SFPQ, RBM38, RBM24, PTB, SRSF10, MBNL1, BRUNOL6, and QKI in this analysis. The enrichment of these motifs suggests potential co-regulation of AS in some of these regulated events by these splicing factors.

The Esrp1 binding site (UGGUGG) was used in the known RBP motif analysis, however, we previously identified the top 6-mer Esrp1 binding motifs by SELEX-Seq, which highlighted the affinity of Esrp1 binding for GU-rich sequences (*Dittmar et al., 2012*). Using all possible 6-mers we confirmed the presence of the Esrp1 UGGUGG motif as described above, but we also observed a growing list of GU-rich sequences including the GGUGGU, GUGGUG, and other UGG-containing motifs identified by SELEX-Seq (*Figure 5—source data 3*). We also determined an Esrp RNA binding map for Esrp regulated SE splicing events using the top 12 Esrp1 binding sites identified by SELEX-Seq. We mapped the enrichment of these motifs in the exon body and in 250 bp intronic sequences flanking the regulated exon and up- and downstream exons (*Figure 5C*). There was a strong enrichment for these Esrp motifs (red line) in the upstream intron of silenced exons in all three comparisons, and enrichment in the exon body of silenced exons with the strongest enrichment in the *Esrp1* KO sensitive exons. Moreover, there was also enrichment of the Esrp motifs in the downstream introns of Esrp enhanced exons (blue line), with the most striking enrichment in the KO. We cannot rule out the

possibility that other changes in splicing factors at the level of AS, localization, or translational efficiency may affect their activity independent of changes in total RNA expression, and in turn alter the regulatory landscape of Esrp responsive splicing events. However, the differential expression of the Esrps together with the enrichment of Esrp binding motifs and their positional enrichment relative to the regulated exons further suggest that many or most of the identified splicing events are likely direct targets.

## *Esrp1/2* DKO epidermis demonstrates large scale changes in gene expression, including components of the epidermal differentiation complex (EDC)

We evaluated changes in total transcript expression levels to further investigate how a loss of Esrp regulated splicing might directly (e.g., *Grhl1*) or indirectly impact global gene expression and contribute to the developmental defects seen in the epidermis of Esrp DKO mice. We performed differentially expressed gene (DEG) analysis on all RNA-seq data generated from the four genetic groups. We identified 572 significant changes in gene expression, with >twofold difference in gene expression and FDR < 5% based on average FPKM in at least one pairwise comparison of the four genetic groupings (*Figure 6A*, *Figure 6—source data 1*). The differential gene expression analysis revealed that by far the most predominant changes in expression are observed between WT and DKO embryonic epidermis. This observation is consistent with the emergence of an epidermal phenotype only in the DKO epidermis. In the WT vs DKO group there were 514 significant changes in gene expression, of which 65% (337/514) were down regulated and 35% (178/514) were up regulated in the DKO epidermis. To identify biological functions of these gene expression changes we utilized GO analysis, using DAVID 6.7, on both the up and down regulated genes from the DKO epidermis. The DEGs were enriched for the GO terms 'epidermal cell differentiation', 'keratinocyte differentiation',

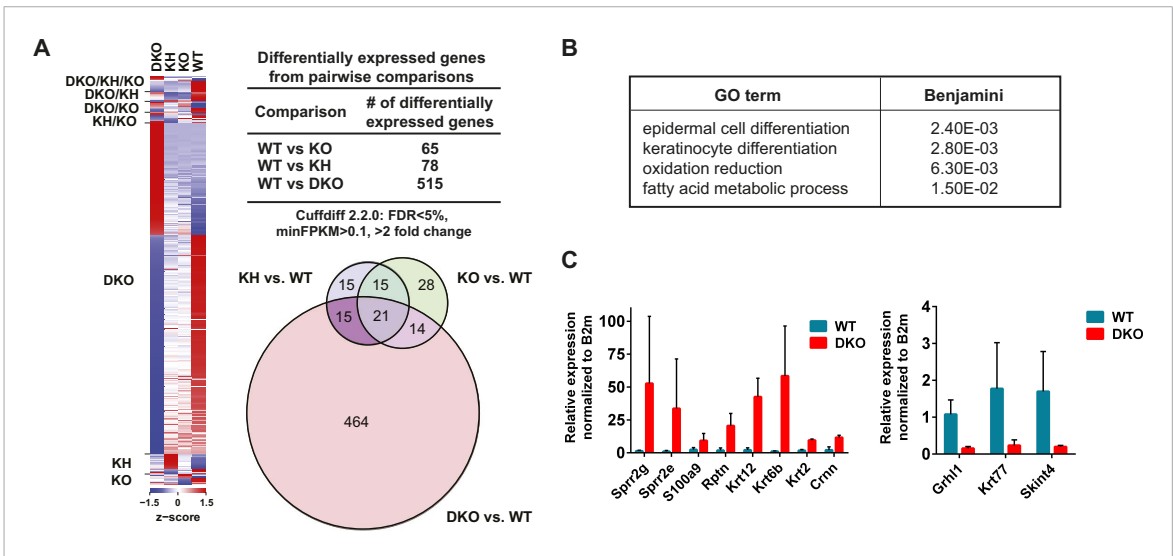

**Figure 6**. Global analysis of differential gene expression in Esrp KO epidermis. (**A**) Heatmap of differentially expressed genes (DEGs) in Esrp KO epidermis. Table of the total number of DEGs from WT comparisons to KO, KH, and DKO samples. Venn diagram depicting the overlap in DEGs from WT comparisons to KO, KH, and DKO samples. (**B**) Functional gene ontology (GO) term enrichment of DEGs from the WT vs DKO comparison (indicated FDR [Benjamini-Hochberg] <0.05 as calculated by DAVID 6.7 are listed). (**C**) qRT-PCR validation of eight epidermal differentiation complex (EDC) genes with increased expression in DKO samples, and down regulation of three genes including the AS-mediated non-sense mediated decay (AS-NMD) target, *Grhl1* (n = 3).

The following source data are available for figure 6:

**Source data 1**. DEG analysis in Esrp deficient epidermis.

**Source data 2**. DAVID analysis of DEGs in DKO epidermis.

'oxidation reduction', and 'fatty acid metabolic process' (FDR [Benjamini-Hochberg] <0.05) (*Figure 6B*, *Figure 6—source data 2*). Similarly, several interesting functional GO terms were associated with p-values <0.01 while not reaching the threshold for multiple comparisons included terms associated with lipid metabolism, epithelial and epidermal differentiation and development, cytoskeletal organization, calcium ion binding, and defense response.

The keratinization and epithelial cell differentiation/development classifications are associated with several families of genes that comprise the EDC (*Patel et al., 2003*; *Kypriotou et al., 2012*). In the DKO epidermis we saw up regulation of several EDC genes, including the *Sprr (1a, 1b, 2b, 2d, 2e, 2g, 2h)*, *Lce (3b, 3c, 3f)*, *S100 (a4, a6, a9)*, and *Repetin (Rptn)* and *Cornelin (Crnn)* genes. We also observed up regulation of epidermal differentiation associated genes including numerous *Keratins (2, 6a, 6b, 12, 13, 16, 23, 34, 84)*. We confirmed increased expression of EDC genes (*Sprr2g*, *Sprr2e*, *S100a9*, *Rptn*, and *Crnn*) and *Keratins* (*Krt12*, *Krt6b*, *Krt2*) by real-time RT-PCR in DKO compared to WT epidermis. Reduced expression of *Grhl1*, *Krt77*, and *Skint4* was also confirmed by real-time RT-PCR (8/11 of tested genes, 73% validation) (*Figure 6C*). Up regulation of EDC genes as well as changes in other genes are often associated with mouse models harboring skin barrier defects, such as *Loricrin*, *Klf4*, and *Grhl3* KO mouse models (*Segre et al., 1999*; *Koch et al., 2000*; *Yu et al., 2006*). While the mechanism of up regulating EDC gene expression is not well understood, it is believed to be a compensatory response to stress or breakdown of the skin barrier function. Because very similar changes in gene expression are observed in *Esrp1/Esrp2* DKO epidermis as those in other mouse KO models associated with barrier defects, we believe these changes in EDC genes and *Keratins* are mostly an indirect effect of the Esrp KO, in contrast to the direct regulation of *Grhl1* by AS-NMD. However, it is possible that some of the changes in gene expression may be due to direct regulation by the Esrps at other post-transcriptional steps, such as mRNA stability or translation. These collective changes in gene expression suggest epithelial barrier dysfunction in the epidermis associated with the loss of the Esrps.

## Conditional KO of the Esrps in the epidermis result in skin barrier defects

The increase in EDC gene expression in the epidermis of DKO embryos suggested a possible defect in skin barrier function. However, as the *Esrp1/Esrp2* DKO mice are not viable at birth we were unable to analyze postnatal barrier defects in DKO mice. Therefore, we utilized a conditional KO approach using transgenic *Keratin14-Cre* (*Krt14-Cre*) mice (*Andl et al., 2004*). This *Krt14-Cre* transgene is first expressed at E11.5 in surface epithelium and maintains expression in basal layer keratinocytes of stratified epidermis (*Zhang et al., 2008*). Because basal keratinocytes undergo differentiation to give rise to all epithelial cells of the interfollicular epidermis, Krt14-Cre effectively induces KO in all epithelial cell layers of stratified epidermis. We generated control mice harboring at least one WT *Esrp1* allele (*Esrp1*$^{flox/+}$, *Esrp2*$^{-/-}$, *Krt14-Cre* Tg– or Tg+) and littermate Krt14-DKO (*Esrp1*$^{flox/flox}$, *Esrp2*$^{-/-}$, *Krt14-Cre* Tg+) mice. Krt14-DKO neonates were viable at birth (P0.5), however no Krt14-DKO pups with confirmed *Esrp1* KO survived to the following day (P1.5). While the early neonatal lethality in the Krt14-DKO mice precluded long term evaluation of barrier function, we were able to evaluate skin barrier function by means of a standard water loss assay, which measures the ability of a neonatal mouse epidermis to retain fluid and prevent dehydration by loss of water through the skin (*Gladden et al., 2010*). Litters of conditional KO neonates were isolated from their mother at P0, and weighed every 30 min for 5 hr. During the time-course control littermates lost weight at expected levels, but never exceeded ~1.5% of total body weight, indicative of an intact epidermal barrier. However, Krt14-DKO pups lost significantly more weight (~2.5–3% of total body weight) compared to littermate controls during the same time period (*Figure 7E*). The *Krt14-Cre* DKO mice that exhibited weight loss during the water loss assay also displayed skin phenotypes observed in other mouse KO models associated with skin barrier defects: thin red shinny skin with areas of dry flaky skin (*Figure 7A*) (*Koch et al., 2000*; *Gladden et al., 2010*).

To confirm the efficiency of *Esrp1* KO with the phenotype in our Krt14-DKO mice, we isolated total epidermis following the water loss assay. RNA from total epidermis was isolated and subject to real-time RT-PCR for *Esrp1* expression. We confirmed an average of ~85% loss of *Esrp1* in our Krt14-DKO P0 pups compared to littermate controls (*Esrp1*$^{flox/+}$, *Esrp2*$^{-/-}$, *Krt14-Cre* Tg–) (*Figure 7B*). We also confirmed Esrp1 protein KO in isolated total epidermis at P0 by western (*Figure 7C*). We confirmed the conditional *Esrp1* KO was sufficient to mediate splicing changes in control compared to

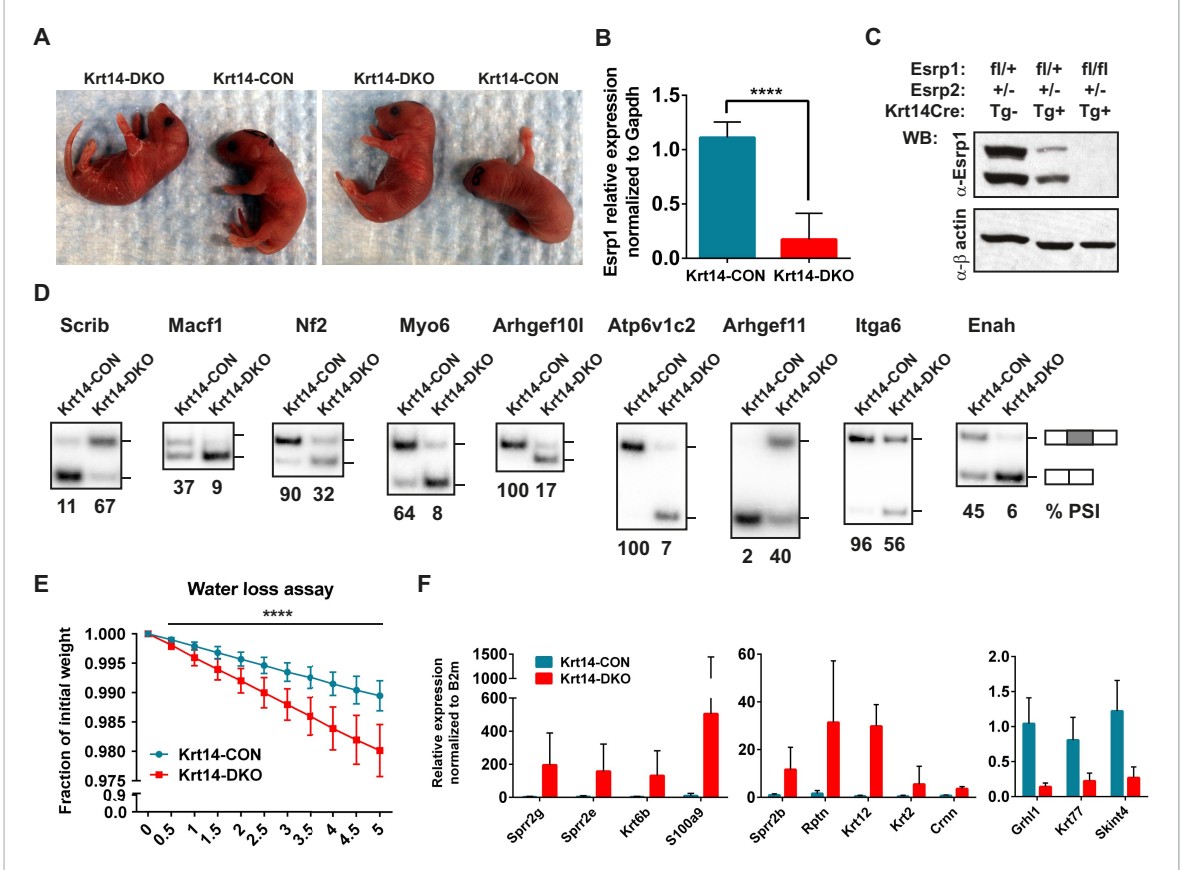

**Figure 7**. Conditional KO of the Esrps in the epidermis result in skin barrier defects. (**A**) Images of P0.5 Krt14-CON and Krt14-DKO pups, which present with dry flaky skin or red shinny skin. (**B**) qRT-PCR validation of Keratin14-Cre mediated KO of the floxed *Esrp1* targeted cassette in isolated Krt14-CON (n = 5) and Krt14-DKO (n = 6) epidermis. (**C**) Western validation of Esrp1 protein KO in Krt14-Cre mice harboring floxed *Esrp1* alleles (n = 2). (**D**) Hot RT-PCR gel images of Esrp SE targets, confirming Krt14-DKO epidermis have defects in Esrp-regulated splicing targets (n = 2). (**E**) Water loss assay for barrier defect in P0.5 pups. Fraction of initial weight is graphed over 5 hr in 30 min intervals. Multiple t-tests analysis indicate significant differences between the Krt14-CON (n = 16) and Krt14-DKO (n = 7) pups. (**F**) qRT-PCR validation of nine EDC genes identified with increased expression in DKO samples, and down regulation of three genes including the AS-NMD target, *Grhl1* (n = 4).

Krt14-DKO epidermis (*Figure 7D*). To determine if the Krt14-DKO also presented with the molecular signature for barrier defect observed in the DKO epidermis, we performed qRT-PCR for the EDC genes and *Keratins* tested in the DKO epidermis. Similar to the DKO epidermis we saw increased expression of the same EDC genes with the addition of *Sprr2b*, and we also confirmed reduced expression of *Grhl1*, *Krt77*, and *Skint4* (9/11 of tested genes, 82% validation) (*Figure 7F*). Therefore, the Krt14-DKO mice share the differential expression of EDC and *Keratin* genes, consistent with the observed skin barrier defect. Taken together these observations implicate the Esrps in the regulation of skin barrier function, a fundamental characteristic of epithelial cells in the epidermis. However, the maintenance of barrier function across epithelial cell layers is a fundamental property of epithelia in many organs throughout the body. Therefore, the Esrp-directed splicing network generates epithelial specific or enriched protein isoforms that are likely to be required for epithelial cell barrier function in epithelial cells throughout mammalian organisms.

## Discussion

This study represents the first in vivo analysis of the epithelial-specific splicing factors, Esrp1 and Esrp2 and the broad roles they play in epithelial cells in a multitude of organs. The numerous developmental defects we identified in these mice demonstrate an essential role for Esrp-regulated splicing in the

development of many organ systems. Using germline KOs for *Esrp1* and *Esrp2* we demonstrate a requirement for Esrp1 during embryonic development and postnatal survival, with loss of both proteins manifesting the most severe phenotypes. To evaluate Esrp regulated splicing in the context of the development of a specific tissue we used the readily accessible epidermis as our model system. We uncovered variable sensitivity of AS events to loss of Esrp1 alone or in combination with loss of Esrp2, which were also enriched for Esrp binding motifs. Global changes in gene expression accompanying loss of the Esrps in the skin of the germline DKO mice revealed an epidermal barrier defect that was confirmed using conditional ablation of the Esrps in the epithelial cells that constitute the epidermis. Together, these observations provide strong evidence for the cell autonomous requirement of the Esrps in epidermal development and programs of epithelial-specific splicing in organogenesis and epithelial cell function.

The characterization of Esrp deficient embryos revealed striking defects in several organ systems. The primary and most readily apparent defect in mice with ablation of *Esrp1* alone was fully penetrant bilateral cleft lip and palate (CL/P). Orofacial defects are amongst the most common forms of birth defects but the causes of CL/P are heterogeneous, including syndromic and non-syndromic genetic disorders and environmental factors (*Dixon et al., 2011*). Importantly, whereas there are greater numbers of patients that have cleft lip with or without cleft palate (CL/P) compared to isolated cleft palate (Cleft Palate Only [CPO]), there are few mouse models for CL/P compared to CPO. The limited mouse models for CL/P include KO models for genes regulating signaling and transcription (*Juriloff and Harris, 2008*). Therefore, the Esrp KO mouse is a unique model that can be used to further evaluate the role of epithelial-specific splicing regulation in lip and palate formation. Based on previous evidence that cell or tissue-specific splicing factors regulate biologically coherent sets of targets genes in the cells that express them, further determination of epithelial specific splice variants could provide molecular insights into novel genes underlying the pathogenesis of CL/P. The orofacial defects were more severe in mice with combined loss of Esrp2. Similarly, mice deficient for Esrp1 and Esrp2 exhibit progressively more severe phenotypes in other organs and tissues, including forelimb, salivary gland, lung, and skin. The functional redundancy of the Esrps during development is consistent with our in vitro observations at the level of AS. Therefore, the progression of phenotypes could be due to the degree of change in AS mediated by loss of Esrp1 alone or in combination with Esrp2.

To evaluate the potential for Esrp1 and Esrp2 co-regulation of AS, we used the epidermis as our model system to identify the global Esrp regulated splicing program in vivo. We illustrated the possible biological impact of varied sensitivity of Esrp targets by highlighting the splicing of exons *-IIIb* and *-IIIc* in the *Fgfr* family of receptors. This splicing event regulates ligand binding specificity and the Fgf signaling axis is a crucial signaling pathway utilized during development in a multitude of organ systems. It is of note that despite the fact that the Esrps are required for expression of Fgfr2-IIIb, *Esrp* DKO embryos do not fully phenocopy the defects previously seen in *Fgfr2-IIIb* KO mice. This is most evident in the craniofacial and limb defects, where KO of *Esrp1* alone results in bilateral cleft lip and palate and DKO embryos have forelimb defects, whereas *Fgfr2-IIIb* KO mice exhibit CPO and absence of both forelimbs and hindlimbs (*De Moerlooze et al., 2000*; *Rice et al., 2004*). Part of this discrepancy can likely be accounted for by differences in the manner in which *Fgfr2* splicing is altered in each model. In our mice, there is a switch in splicing from exon IIIb to IIIc in *Esrp* DKO mice that would still encode a functional receptor, albeit one with altered ligand binding specificities. In contrast, the deletion of exon IIIb in *Fgfr2-IIIb* KO mice does not result in default splicing to exon IIIc in epithelial cells, but instead results primarily in skipping of both exons IIIb and IIIc, thereby inactivating *Fgfr2* expression altogether in epithelial cells, but retaining Fgfr2-IIIc in mesenchymal cells. Hence in some contexts, autocrine or paracrine functions of Fgf ligands for Fgfr2-IIIc may compensate for the switch in *Fgfr2* splicing, but not in others. For example, ablation of Fgfr2-IIIb and its mesenchymal derived ligand Fgf10, result in failure to form lungs and submandibular (salivary) glands, indicating that the switch in Fgfr2 isoforms in *Esrp* DKO mice may be sufficient to phenocopy these defects, but not others. However, it must also be emphasized that Esrp ablation also induces global changes in AS that surely contribute to the phenotypes we observe, including those, such as cleft lip, that were not identified in Fgfr2-IIIb KO mice. In the epidermis, we suspect that alterations in Fgf receptor splicing and Fgf signaling contribute to the phenotypes we observed since epidermal hypoplasia was observed in both Fgfr2-IIIb and *Esrp1/Esrp2* DKO skin. However, conditional KO of Fgfr2-IIIb as well as combined KO of Fgfr2 and Fgfr1-IIIb in the epidermis is not lethal unlike the post-natal lethality we

observe when *Esrp1* and *Esrp2* are conditionally ablated in epidermis. While skin from mice with epidermal-specific ablation of *Fgfr2-IIIb* or *Fgfr2/Fgfr1-IIIb* exhibits some inflammation and/or a later onset barrier defect in adult skin, these mice survive for at least several months post-natally. These observations, combined with the fact that *Esrp1/Esrp2* ablation in the epidermis is associated with an isoform switch rather than complete loss of function of Fgfr1 and Fgfr2 receptor proteins, strongly suggest that splicing changes in additional Esrp target transcripts are required for the lethal barrier defect we observe. Therefore, the challenge for future studies will be to identify the individual or combined splicing events that underlie the observed phenotypes in the skin as well as other tissues and organs.

In many of the Esrp targets we identified large 'switch-like' changes in AS with the loss of the Esrps. These changes would lead to an altered proteomic landscape by the loss or introduction of different protein isoforms. For example, *Fgfr2* undergoes a full switch from *-IIIb* to *-IIIc*, thus entirely swapping one isoform for its alternative isoform. We also observed cases where the loss of the Esrps resulted in the loss of an isoform or the inverse where a new isoform was introduced (e.g.,: Fgfrs, Enah, Arhgef11, Itga6, Macf1, Nf2, etc). These changes in protein isoforms likely underlie the observed phenotypes in the Esrp deficient mice. The variable sensitivity of AS events to the loss of Esrp1 vs combined loss of Esrp1 and Esrp2 was not restricted to a handful of Esrp targets, but rather was observed in the global splicing analysis in the epidermis. The fact that Esrp2 can only partially compensate for loss of Esrp1 in the epidermis, despite retaining high expression in Esrp1 KO mice, indicates that this differential sensitivity partly reflects reduced splicing activity of Esrp2 compared to Esrp1. It is important to emphasize that the levels of both *Esrp1* and *Esrp2* expression are high in the epidermis (Average FPKM: Esrp1 130, Esrp2 127), which in turn could impart greater functional redundancy in the epidermis. However, not all epithelial cells express *Esrp1* or *Esrp2* as highly as in the skin and most epithelial cells express less *Esrp2* relative to *Esrp1* (*Su et al., 2002*). The differences in phenotypes observed in the *Esrp1* KO and *Esrp1/Esrp2* DKO may therefore reflect splicing alteration in specific transcripts for which Esrp2 cannot fully compensate for loss of Esrp1 and/or be reflective of differences in Esrp2 levels in different epithelia.

To determine a potential mechanism underlying the varied AS sensitivity we mapped Esrp RNA binding motifs for the SE cassette exons using the GU-rich sequences identified by SELEX-Seq (*Dittmar et al., 2012*). The SE events identified from the KO epidermis were enriched for Esrp binding motifs in the upstream intron and exon body of silenced exons, and within the downstream intron of enhanced exons proximal to the 5′ splice site. This RNA binding map is consistent with strong repression and enhancement, respectively. This motif enrichment pattern is most strikingly observed in the KO dataset, as the exon body enrichment is reduced/lost in the KH and DKO samples. This describes a potential mechanism in which the *Esrp1* KO sensitive AS events require more Esrp binding sites for maintenance of the Esrp-directed splicing pattern. In the absence of Esrp1 expression, Esrp2 is not sufficient to drive AS regulation or to block the activity of other splicing factors that promote the opposite pattern of splicing. Despite the absence of major changes in gene expression of other splicing factors, it is likely that other splicing factors contribute to the regulation of Esrp AS targets. RBFOX2 has been shown to co-regulate ESRP AS events, including those associated with EMT, despite only small changes in RBFOX2 expression in EMT (*Dittmar et al., 2012*). Similarly the ubiquitously expressed hnRNPM was shown to regulate splicing of *CD44* during EMT, but only in conjunction with down regulation of the Esrps (*Xu et al., 2014*).

The Esrps direct a program of AS that is highly conserved between human and mouse and enriched in biologically coherent pathways involved in epithelial cell biology and function. The Esrps are also highly conserved in the three RRMs in the orthologous Fusilli in fruit flies (*Drosophila melanogaster*) and Sym-2 in round worm (*Caenorhabditis elegans*). We have shown that Fusilli is partially functional in directing ESRP splicing events when transfected in human cells lacking the ESRPs, indicating clear conservation of function (*Warzecha et al., 2009a*). The expression of Fusilli is also epithelial-specific and both Fusilli and Sym-2 have been shown to function as a regulator of endogenous AS in *Drosophila* cells and *C. elegans*, respectively (*Wakabayashi-Ito et al., 2001*; *Barberan-Soler and Zahler, 2008*) (unpublished data). In *Drosophila*, Fusilli homozygous mutation was lethal, indicating that Esrp and Fusilli are both essential splicing factors despite substantial phylogenetic distance between mammals and flies. Although the genomic architecture of fly and worm genes differ greatly from higher eukaryotes,

precluding direct comparison of splicing events, it will be interesting to further explore whether these evolutionarily conserved orthologues of the Esrps also regulate programs of splicing that function in similar processes and pathways.

The epidermis served as a model to evaluate epithelial cell function, as the skin serves as the primary defense to the external environment as well as maintaining fluid homeostasis. The increase in EDC gene expression in the DKO epidermis is likely an indirect effect of the loss of the Esrps, but in turn it implicates Esrp regulated splicing in the maintenance of epithelial cell function necessary for barrier function in vivo. For example, we confirmed the differential splicing of several genes that have been evaluated in KO mice for skin barrier function and maturation of cellular junctions. *Nf2* (*Merlin*) exhibited a 'switch-like' change in AS in the DKO epidermis, transitioning from nearly full inclusion to full skipping (*Figure 4B*). Similar to our Krt14-DKO mice, conditional ablation of *Nf2* using Krt14-Cre resulted in mice with dry, flaky skin, and epidermal barrier defects as measured by water loss assay. Biochemical evaluation identified Nf2 as an interaction partner with the adherens junction (AJ) and Par3 necessary for the maturation of AJ to mature TJs (*Gladden et al., 2010*). While this analysis was performed in complete KO of *Nf2*, evaluating the isoform specific functions of Nf2 under regulation by the Esrps could provide further insight into the maturation of junctional complexes required for epithelial cells and barrier function. The Esrps directly regulate the levels of *Grhl1* by promoting exon 5 inclusion and preventing transcript degradation by NMD. Mice lacking Grhl1 have reduced expression of desmosomal cadherins resulting in mild anchoring defects of hair follicles in the skin (*Wilanowski et al., 2008*). Epidermal barrier defects by way of epidermal hyperplasia and inflammatory cell infiltration were observed in adult mice lacking Grhl1 (*Mlacki et al., 2014*). *Cd44* has a complex pattern of splicing in which the variable region encoded by 10 exons is included in epithelial cells, and skipping of these exons is observed in non-epithelial cells. Mice lacking Cd44, which in the case of the epidermis is the Esrp regulated epithelial form Cd44v, have delays in outside-in barrier formation during embryonic development and barrier formation measured in cultured keratinocytes (*Kirschner et al., 2011*). These examples of Esrp splicing targets involved in skin barrier function highlight the roles of the Esrps in regulating known genes involved in this process. Defects in epithelial barrier function are associated with human disorders in multiple organs including atopic dermatitis, allergen response in respiratory tracts, and gastrointestinal inflammation (*Segre, 2006*). We predict that numerous Esrp-regulated targets will also be shown to play roles in epithelial barrier function in the skin and other epithelial populations. Thus, the *Esrp* DKO mice can be used to evaluate isoform-specific functions of known barrier genes, as well as identify novel genes involved in the maintenance of epithelial barrier function via functions of epithelial specific protein isoforms.

The KO of the Esrp proteins in mice provides the first evidence for a requirement for these cell-type-specific splicing factors in embryonic development and epithelial cell functions. The identification of the Esrp regulated splicing program provides numerous examples of genes that merit further investigation into their developmental functions and contribution to human diseases such as CL/P and diseases associated with epithelial barrier defects. A challenge for future studies will be to functionally dissect the molecular mechanisms by which some of these isoform-specific activities are involved in maintenance not only of epithelial barriers, but also in mediating epithelial-mesenchymal crosstalk during development and maintaining other properties of epithelial cells. The compilation of Esrp regulated in vivo splicing targets in tissue-specific epithelial populations will provide further insights into the posttranscriptional splicing networks in epithelial cells at play during development and organogenesis of various tissues.

## Materials and methods

### Generation of Esrp KO mice

The targeting vector for *Esrp1* was designed in collaboration with the Penn Gene Targeting Core (*Figure 1—figure supplement 2A*). The vector features a loxP site in intron 6 upstream of exon 7, a neomycin selection cassette flanked by FRT sites in intron 9 followed by a loxP site. The targeting vector was electroporated into V6.5, hybrid C57BL6/129Sv, mouse ES cells and neomycin resistant clones were isolated. Genomic DNA was prepared from the ES clones and used for screening by PCR and Southern blot using standard techniques. For Southern blot analysis genomic DNA was digested with RE ScaI for 5′ probe B or HpaI for 3′ probe C, gel electrophoresed and transferred to nitrocellulose

membranes. Membranes were probed using 32-P-dCTP labeled probes targeting intron 4 (5′ end) or intron 11 (3′ end) outside of the arms of homology. Clone 1D1 was selected (*Figure 1—figure supplement 2B*), and injected into Balb/c blastocysts and implanted in pseudo-pregnant females. 21 total chimeras were generated, from which males were crossed to C57BL/6J females. Germline transmission was confirmed by PCR and Southern blot as described above. The resulting germline mice served as the basis for both the conditional and KO alleles. The conditional allele was generated by crossing of male #114, from chimera 1.9, to B6(C3)-Tg(Pgk1-FLPo)10Sykr/J females to remove the neomycin cassette. Offspring were screened for removal of the neomycin cassette by PCR analysis. Heterozygous *Esrp1*$^{flox/+}$ mice were crossed to generate homozygous *Esrp1*$^{flox/flox}$ conditional mice, with no apparent phenotype, and to maintain the line. The Esrp1 KO allele for ubiquitous ablation of Esrp1 was generated by crossing of male #31, from chimera 1.10, to C57BL/6-Tg(Zp3-cre)93Knw/J females. Resulting offspring were screened by PCR and DNA sequencing for Cre-mediated recombination resulting in the complete deletion of neomycin cassette and the genomic region encoding exons 7–9. *Esrp1*$^{+/-}$ heterozygotes were crossed to generate homozygous *Esrp1*$^{-/-}$ KO mice and line maintenance.

The Esrp2 KO allele was generated as part of the KOMP and purchased from Velocigene (C57BL/6N-Esrp2tml(KOMP)$^{vlcg}$). The Esrp2 gene locus was replaced by a LacZ and floxed neomycin selection cassette. LacZ expression would be placed under the Esrp2 promoter, while the neomycin expression is driven by the human ubiquitin C promoter. The C57BL6 derived *Esrp2* KO ES clone, AG3, was injected in to Balb/c blastocyctes. 14 total chimeras were generated and male chimeras were crossed to C57BL/6J females. Germline transmission was confirmed by PCR. Heterozygous *Esrp2*$^{+/-}$ mice were crossed to generate *Esrp2*$^{-/-}$ KO mice, which present with no overt phenotype.

## In situ hybridization

Embryonic and adult in situ hybridization analysis of *Esrp2* was performed by Phylogeny (Columbus, Ohio) as previously described for *Esrp1* in *Warzecha et al. (2009a)*. Primers to generate the 675 bp probe to *Esrp2* were (forward) 5′-GAAGCTT<u>CTCACCACCTACCTACGCCACCTT</u>-3′ and (reverse) 5′-GGAATTC<u>AGCAGCCCTATCACATCTCCAG</u>-3′. PCR products were cloned into the pDP19 vector (Ambion) for in vitro transcription. For *Esrp1* in situ analysis dorsal skin was removed from P0 embryos following euthanasia and fixed for 4 hr in 4% paraformaldehyde then washed in PBS. Skin samples were cryoprotected overnight in 30% sucrose/PBS then snap frozen in OCT embedding compound (Sakura Finetek Torrence, CA). Frozen skin was sectioned at 20 μm and hybridized with digoxigenin-UTP-labeled riboprobes against *Esrp1* (*Warzecha et al., 2009a*). In situ hybridization was performed as previously described (*Nissim et al., 2007*).

## Mouse crosses

*Esrp1* and *Esrp2* ubiquitous KO embryos for phenotype and molecular analysis were generated by three crossing schemes: *Esrp1*$^{+/-}$, Esrp2$^{+/+}$ to *Esrp1*$^{+/-}$, Esrp2$^{+/+}$ mice for *Esrp1*$^{-/-}$, Esrp2$^{+/+}$ (E1 KO, KO) mice; *Esrp1*$^{+/-}$, Esrp2$^{+/+}$ to *Esrp1*$^{+/-}$, *Esrp2*$^{-/-}$ mice for *Esrp1*$^{-/-}$, Esrp2$^{+/-}$ (KO/Het, KH) mice; and *Esrp1*$^{+/-}$, *Esrp2*$^{-/-}$ to *Esrp1*$^{+/-}$, *Esrp2*$^{-/-}$ mice for *Esrp1*$^{-/-}$, *Esrp2*$^{-/-}$ (DKO) mice. Control litter mates have at least one wild-type allele of *Esrp1*. Timed matings were performed as per standard procedures. Pregnant dames were CO2 euthanized, and embryos were isolated and euthanized by decapitation or cryoeuthanasia where whole intact embryos were needed for histology. Krt14-Cre mice (*Andl et al., 2004*) provided by Dr Sarah Millar were crossed to *Esrp1*$^{flox/flox}$, *Esrp2*$^{-/-}$ mice to generate breeder males, *Esrp1*$^{flox/+}$, *Esrp2*$^{-/-}$, Krt14-Cre Tg+. The breeder males were then crossed to *Esrp1*$^{flox/flox}$, *Esrp2*$^{-/-}$ to generate the Krt14-DKO mice, *Esrp1*$^{flox/flox}$, *Esrp2*$^{-/-}$, Krt14-Cre Tg+. All animal procedures and experiments were approved by the Institutional Animal Care and Use Committee (IACUC) at the University of Pennsylvania.

## Histological and gross morphology analysis of Esrp KO mice

Whole embryos or isolated tissues were fixed in 4% Paraformaldehyde overnight at 4°C, followed by PBS washes and transfer to 70% ethanol for processing and paraffin embedding for sectioning. H&E stains were performed for gross histological analysis. Measurements of skin for epidermal thickness, hair foillicle number and follicle stage were performed using an Olympus BX43 and CellSens software. E18.5 embryos were isolated weighed, measured from crown to rump, and photographed. One-way-ANOVA

multiple comparisons analysis was performed, adjusted p-values for multiple comparisons (*) $p < 0.05$, (**) $p < 0.01$, (***) $p < 0.001$, (****) $p < 0.0001$.

## Bone and cartilage stains

Alcian Blue/Alizarin Red staining of bone and cartilage was performed as in *Ovchinnikov (2009)* with minor modifications. Briefly, euthanized embryos were placed in tap water overnight at 4°C. Embryos were placed in 70°C water for 20–30 s allowing easy removal of skin. Visceral organs were removed and the embryos were placed in 95% ethanol for 5 days, acetone for 2–4 days, then in staining solution (1/20–0.3% Alcian Blue 8GS in 70% ethanol, 1/20–0.1% alizarin red S in 70% ethanol, 1/20- glacial acetic acid, 17/20 70% ethanol) for 3–4 days. Embryos were cleared in 1% KOH followed by 20%, 50%, and 80% glycerol in 1% KOH. Stained embryos were stored in 100% glycerol and imaged using an inverted Olympus dissecting microscope.

## Water loss assay

The water loss assay was performed as in *Gladden et al. (2010)*, with minor modifications. In short, P0 pups were removed from their cage and maintained at 39°C to approximate maternal thermal regulation. Pups were numbered and weighted at time zero and every 30 min thereafter for 5 hr. Pups were then euthanized to isolate epidermis to confirm KO efficiency and molecular analysis (see 'Isolation of epidermis for RNA and protein analysis').

## Isolation of epidermis for RNA and protein analysis

E18.5 embryos were isolated and cryoeuthanized on ice for 20 min, P0 Krt14 pups were $CO_2$ euthanized then cryoeuthanized on ice for 20 min. Embryos were processed through two 70% washes and two washes in HBSS + Penicillin/Streptomycin (P/S) for 2 min each. Head, limbs, and tail were removed followed by incision along the dorsal A/P midline. Trunk skin was removed and floated dermis side down on 0.25% trypsin/HBSS at 4°C for 16–18 hr. Epidermis and dermis were manually separated using forceps, rinsed in 10% FBS/HBSS then in HBSS + P/S and snap frozen on liquid nitrogen. Epidermis was lysed for RNA isolation in Trizol (Invitrogen, Carlsbad, CA) and resuspended in 10 mM Tris pH 8.0. Protein was isolated in RIPA buffer (Santa Cruz, Dallas, TX).

## RT-PCR and real-time RT-PCR

For synthesis of cDNA 1ug of total epidermal RNA was used for epidermal samples using random hexamer primed M-MLV reverse transcriptase (Promega, Madison, WI). Real-time analysis of Esrp expression was evaluated using Taqman probes for Esrp1 (Mm01220936_g1), Esrp2 (Mm00616290_m1), and Gapdh (Mm99999915_g1) (LifeTechnologies) using a 7500 Fast Realtime machine (AppliedBiosystems). Semi-quantitative radioactive RT-PCR products were separated on 5% PAGE gels, dried and exposed on phosphorscreens, scanned on a Typhoon FLA 9500. Semi-quantitative RT-PCRs for *Exoc1*, *Golga2*, and *Stx2* were separated on 2.5% agarose gels, stained with ethidium bromide, and scanned on a Typhoon FLA 9500. PCR products were quantified using ImageQuant TL, version 7.0. Splicing ratios are represented as PSI for cassette exons and were normalized to RT-PCR product size. Quantification of exon IIIb and IIIc for *Fgfr1*, *Fgfr2*, and *Fgfr3* required restriction enzyme specific to discriminate the two isoforms. *Fgfr2* PCR products were digested with AvaI (IIIb) or HincII (IIIc). *Fgfr1* products were digested with BstXI (IIIb) and HincII (IIIc). *Fgfr3* products were digested with StuI (IIIb) and PstI (IIIc) (all restriction digestions were performed according to NEB guidelines at 5 U/digestion). Primer sequences are listed in *Supplementary file 3*. Graphical representation of PSI for IIIb inclusion were calculated as the ratio of IIIb/(IIIb + IIIc).

## Western blot and immunofluorescence

Protein lysate from total epidermis was used for detection of Esrp1 and Esrp2. 10 µg of protein lysate was separated on Bis-Tris 4–20% Gradient SDS-PAGE gels (Invitrogen) and transferred to nitrocellulose membranes. Membranes were blocked in 5% Non-fat Milk in PBST then incubated overnight at 4°C in primary antibody using mouse monoclonal antibody to Esrp1 and Esrp2, 23A7, at 1:1000 dilution (Rockland Immunochemicals). Sheep anti-mIgG:HRP at 1:2500 (GE Healthcare) and ECL detection (Inivtrogen) by chemiluminescence. Loading control using anti-Beta Actin M2 (Sigma) at 1:10,000 and sheep anti-mIgG:HRP at 1:10,000 dilution. 7 µm sections from paraffin embedded dorsal

skin was dewaxed and rehydrated according to traditional proceedures, and antigen retrieval was performed using antigen unmasking solution (Vector Laboratories). Primary antibodies for Keratin14 (AF 64) (Covance, Princeton, NJ), Keratin10 (Covance), Loricrin (AF 62) (Covance), Flaggrin (Covance), p63 (4A4) (Santa Cruz), β-catenin (15B8) (Sigma, St. Louis, MO), and Lef-1 (C18A7) (Cell Signalling) were used at 4° overnight, followed by fluorescent secondary antibodies Goat-anti-rIgG AlexaFlour 488 or Goat-anti-mIgG F(ab′)2 594 (Invitrogen). Images were taken using an Olympus BX43.

## RNA sequencing and data analysis

Total RNA from individual E18.5 mouse embryo skins were used for 2–3 independent biological replicates for each genotype. Each RNA sample (1 µg) was used for poly A selected RNA-seq library preparation using the TruSeq Stranded mRNA LT Sample Prep Kit (Illumina, San Diego, CA). Biological replicates were individually barcoded, pooled, and split over 2 lanes of a HiSeq 2000 for 100 × 2 bp paired-end RNA-seq at the Penn Next Generation Sequencing Core (NGSC) Facility. The RNA-seq data has been deposited into the NCBI Gene Expression Omnibus under the accession number GSE64357. We mapped RNA-seq reads to the mouse genome (mm10) and transcriptome (Ensembl, release 72) using the software TopHat (v1.4.1) allowing up to 3 bp mismatches per read and up to 2 bp mismatches per 25 bp seed. We used Cuffdiff (v2.2.0) (*Trapnell et al., 2010*) to calculate RNA-seq based gene expression levels using the FPKM metric (fragments per kilobase of exon per million fragments mapped) then identified differential gene expression between the two cell types at FDR < 5%, >twofold difference in gene expression based on average FPKM, and minFPKM > 0.1. To identify differential AS events between the WT and three knockouts (KO, KH, and DKO), we used rMATS v3.0.8 (http://rnaseq-mats.sourceforge.net) to identify differential AS events from strand-specific RNA-seq data corresponding to all five basic types of AS patterns (see the entire list in *Figure 4—source data 1*). Briefly, rMATS uses a modified version of the generalized linear mixed model to detect differential AS from RNA-seq data with replicates (*Shen et al., 2014*). It accounts for exon-specific sequencing coverage in individual samples as well as variation in exon splicing levels among replicates. For each AS event, we used both the reads mapped to the splice junctions and the reads mapped to the exon body as the input for rMATS. Each KO group was compared to the WT group to identify differentially spliced events with an associated change in PSI (ΔPSI or Δψ) of these events. To compute p-values and FDRs of splicing events with $|\Delta\psi| > 0.01\%$ cutoff, we ran rMATS using -c 0.0001 parameter. The splicing events in *Figure 4—source data 1* summarize detected events with an FDR < 5% and $|\Delta\psi| \geq 5\%$.

## Motif enrichment analysis

We sought to identify binding sites of splicing factors and other RBPs that were significantly enriched in differential exon skipping events between two genetic groups as compared to control (non-regulated) alternative exons. These SE datasets were comprised of 68 (32 enhanced, 36 silenced) in KO, 97 (54 enhanced, 43 silenced) in KH, and 214 (119 enhanced, 95 silenced) in DKO epidermis. We collected 115 known binding sites of RBPs including many well-characterized splicing factors from the literature (*Ule et al., 2003*; *Anderson et al., 2012*; *Dittmar et al., 2012*; *Ray et al., 2013*; *Vanharanta et al., 2014*). For each motif, we scanned for motif occurrences separately in exons or their 250 bp upstream or downstream intronic sequences. For intronic sequences, we excluded the 20 bp sequence within the 3′ splice site and the 6 bp sequence within the 5′ splice site. Alternative exons without splicing changes (rMATS FDR > 50%, maxPSI > 15%, minPSI < 85%) in highly expressed genes (average FPKM > 5.0 in at least one genetic group) were treated as control exons. For each motif, after we counted the number of occurrences in the differentially spliced exons and the control exons, we calculated p-value for motif enrichment via Fisher's exact test (right-sided) and used Benjamini-Hochberg FDR correction to adjust for multiple testing for exons, upstream intronic sequences, and downstream intronic sequences separately. A motif can be counted multiple times in a given sequence (by-NT) or a motif can only be counted zero or one time in a given sequence (by-Sequence). We conducted both approaches and identified enriched motifs at FDR(by-NT) < 5% and p-value(by-Sequence) < 0.01.

## RNA map analysis

We used the top 12 GU-rich Esrp binding sites previously identified by the SELEX-Seq (*Dittmar et al., 2012*) to identify the RNA binding map of the Esrps for the differential SE events between two genetic

groups as compared to control alternative exons. We assigned motif scores based on the overall percentage of nucleotides covered by any of 12 Esrp binding sites within a 50-nt window. We slid this window by 1-nt across the exon body and 250 nt of upstream and downstream intron as well as the upstream and downstream exons.

## Acknowledgements

We thank Jean Richa of the Penn Transgenic and Chimeric Mouse Core and Tobias Raabe of the Penn Gene Targeting Core for their help in generating the mouse lines; Trevor Williams (University of Colorado Denver) for protocol and consultation of bone stains and craniofacial phenotypes; Sarah Millar and Fang Liu (UPenn) for protocol guidance in epidermis isolation, reagents for skin immunofluorescence, and providing the *Krt14-Cre* transgenic mice; and Stephen Prouty of the Skin Disease Research Center for histological processing. We also thank Doug Epstein for critical review of the manuscript and technical support as well as the personnel from the Penn Next Generation Sequencing Core (NGSC) for consultation and RNA-Seq analysis.

## Additional information

### Funding

| Funder | Grant reference | Author |
| --- | --- | --- |
| National Institute of General Medical Sciences (NIGMS) | R01 GM088809 | Russ P Carstens |
| National Institute of Arthritis and Musculoskeletal and Skin Diseases (NIAMS) | R56 AR066741 | Russ P Carstens |
| National Institute of Dental and Craniofacial Research (NIDCR) | R56 DE024749 | Russ P Carstens |
| National Institute of Arthritis and Musculoskeletal and Skin Diseases (NIAMS) | P30 AR057217 (sub-award) | Russ P Carstens |
| National Institute of Arthritis and Musculoskeletal and Skin Diseases (NIAMS) | P30 AR050950 (sub-award) | Russ P Carstens |
| National Institute of Diabetes and Digestive and Kidney Diseases (NIDDK) | P30 DK050306 (subaward) | Russ P Carstens |
| National Institute of Diabetes and Digestive and Kidney Diseases (NIDDK) | T32DK700638 | Thomas W Bebee |
| National Institute of Diabetes and Digestive and Kidney Diseases (NIDDK) | F32DK098917 | Thomas W Bebee |
| National Institute of General Medical Sciences (NIGMS) | R01 GM088342 | Yi Xing |
| National Institute of General Medical Sciences (NIGMS) | R01 GM105431 | Yi Xing |
| Alfred P. Sloan Foundation (A.P. Sloan Foundation) | Fellowship | Yi Xing |

The funders had no role in study design, data collection and interpretation, or the decision to submit the work for publication.

### Author contributions

TWB, JWP, YX, RPC, Conception and design, Acquisition of data, Analysis and interpretation of data, Drafting or revising the article; KIS, AMR, Acquisition of data, Analysis and interpretation of data; CCW, Acquisition of data, Analysis and interpretation of data, Contributed unpublished essential data or reagents; BWC, Acquisition of data, Drafting or revising the article

## Ethics

Animal experimentation: Experiments involving use of animals were conducted at the University of Pennsylvania through the office of University Laboratory Animal Resources (ULAR) that oversees all mouse work and is accredited by the Association for Assessment and Accreditation of Laboratory Animal Care (AAALAC). All mouse work was fully approved by the University of Pennsylvania Institutional Animal Care and Use Committee (IACUC) under protocols #803164 and #804789.

# Additional files

### Supplementary files

• Supplementary file 1. Mapping statistics for the epidermis samples used for rMATS splicing analysis. Epidermis samples and the associated reads from 2 lanes of a HiSeq 2000, $2 \times 100$ bp paired-end reads. Total reads and associated mapped reads when aligned to the mm10 mouse genome. Percent of mapped reads assigned to genomic regions are indicated.

• Supplementary file 2. DAVID analysis of all SE splicing events.

• Supplementary file 3. Primer sequences.

### Major dataset

The following dataset was generated:

| Author(s) | Year | Dataset title | Dataset ID and/or URL | Database, license, and accessibility information |
|---|---|---|---|---|
| Xing Y, Carstens R | 2015 | Knockout mice reveal an essential role for Epithelial splicing regulatory proteins (Esrps) in mammalian development and epithelial splicing in vivo | www.ncbi.nlm.nih.gov/geo/query/acc.cgi?acc=GSE64357 | Publicly available at the NCBI Gene Expression Omnibus (Accession no: GSE64357). |

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
