## [Decision Letter]

Thank you for submitting your work entitled “The splicing regulators Esrp1 and Esrp2 direct an epithelial splicing network essential for mammalian development” for peer review at *eLife*. Your submission has been favorably evaluated by James Manley (Senior editor), and 3 reviewers, one of whom is a member of our board of Reviewing editors. One of the three reviewers, Chris Burge, has agreed to share his identity.

The reviewers have discussed the reviews with one another and the Reviewing editor has drafted this decision to help you prepare a revised submission.

Bebee et al present the phenotypic analysis of mice carrying knockout mutations for the Epithelial Splicing Regulatory Proteins, ESRP1 and ESRP2. These proteins, discovered by the Carstens lab, control important alternative splicing events in epithelial cells, but the biological roles of their regulatory programs is not well understood. The authors generated conditional KO alleles of the two proteins. As expected from earlier studies, the ESRP1 mutation has a more severe phenotype than the ESRP2. They make germline null mutations in each and characterize the developmental defects in the ESRP1 knockout (KO), this KO with a heterozygous ESRP2 mutation (KOH), and the double KO (DKO). These mutations have extensive, pleotropic phenotypes and exhibit neonatal lethality. For the DKO, the authors describe cleft palate and a variety of cranio-facial defects, defects in forelimb development, and a variety of organ defects including failure to develop lung or salivary gland. Some defects are much reduced in the KO or the KOH mice, demonstrating partial redundancy in the function of the ESRP's. Focusing on the epidermis, the authors' document reduced thickness in the DKO and reduced numbers of follicles. Nevertheless marker gene expression indicates that all the appropriate cell types and layers are present. Using whole transcriptome profiling of gene expression and splicing by RNAseq in dissected epidermis, they identify many changes in splicing in the mutant epidermis, including many known ESRP targets. By examining all combinations of heterozygous and homozygous mutations, they demonstrate differing sensitivities of splicing events to the ESRP proteins. Some transcripts exhibit wild type splicing with one copy of either gene present, but then shift their splicing entirely, with the loss of the last copy. Other transcripts show more graded responses with intermediate levels of splicing in the compound heterozygotes. The authors identify changes in other RNA binding proteins in response to the knockout, but since these are rarely more than 1.5 fold, argue that these changes are unlikely to cause the splicing alterations observed. Motif analyses within the affected exon regions shows enrichment of ESRP sites upstream of ESRP exons, and downstream of ESRP enhanced exons, a pattern observed previously for several splicing regulators, and indicating likely direct regulation by ESRP of most of the altered exons. Cuffdiff is used to examine genes that are differentially expressed in the mutant epidermis, with by far the most changes found in the DKO. By Gene Ontology analysis, these genes are strongly enriched in process terms related to keratinization and epidermal differentiation, with the former downregulated and the latter upregulated. Finally, they use a Keratin-Cre line to specifically knockout the proteins in the basal keratinocytes, observing similar gross phenotype in that tissue and similar changes in splicing and expression. They demonstrate a loss of epidermal integrity in these mice using a water loss assay.

This paper presents the first characterization of ESRP genetics and is likely to be a foundation for many additional studies of these proteins. The analysis is thorough and carefully done and represents an important advance in understanding the biology of the ESRP's and of splicing regulatory networks more generally.

Major issues:

1) The study presents no histological data on the normal expression of the ESRPs in the tissues affected, or showing its loss in the knockout. The immunoblots shown do not tell us what cells in those tissues are expressing ESRP1, ESRP2 or both, and when they are expressed. These data are needed to interpret some of these phenotypes, particularly in the more complex genotypes. For example, while reading the manuscript and considering the phenotypes in Figure 1 and Figure 2, the reader is left wondering where ESRP1/2 are expressed. Are the phenotypes developing in bone/limb because their expression is exclusive to these tissues?

2) Figure 1 and Figure 2 appear to constitute something of a separate story from the epidermal studies that come later and where the detailed molecular analysis was performed. Since no molecular analysis was performed in the phenotypic tissues of Figure 1 and Figure 2, it is hard to relate these data to the later analysis. The authors could either perform some of these analyses (which may be more difficult in these tissues, moving these data to a supplement, or somehow relating these early figures to a paper that is otherwise about the epidermis.

3) Wording. The authors use the term “splicing regulatory network” (SRN) to describe their analysis of splicing changes in Esrp1 & Esrp2 KO mice. But the analysis primarily focuses on direct regulatory targets of Esrp1/Esrp2: large changes in the expression of other splicing regulatory factors (SRFs) are not observed, splicing changes in other SRFs were not specifically addressed, nor were other steps taken to infer a network of changes downstream of Esrp perturbation. Therefore, the word “network” does not seem justified and SRN should be replaced by the phrase “splicing regulatory program” or similar.

4) The authors describe the GO analysis of the splicing changes, but show no results. They should add this to the supplement and provide p-values (corrected for multiple comparisons).

5) The results would be greatly strengthened by CLIP-seq analysis of an Esrp protein to confirm direct regulatory targets. The authors are likely generating these data and should add it if possible.

6) The authors spend considerable space showing the splicing of FGFR, and discussing this event as an underlying cause of the phenotype, but do not provide evidence. They should be clearer about what portion of the phenotype observed might be attributable to defects in FGF signaling.

---

## [Author Response]

We were pleased that the reviewers had a generally favorable view of our study, which is the first characterization of Esrp KO mice developed in our lab. We thank the editors and reviewers for their attention to the manuscript and the thoughtful comments. We have carefully reviewed the critiques and suggestions for improving the manuscript and have addressed the major concerns in a modified resubmission. Below we address each of the issues or concerns point by point to clarify how we have made changes in the resubmission to address each critique or suggestion.

*Major issues*:

*1) The study presents no histological data on the normal expression of the ESRPs in the tissues affected, or showing its loss in the knockout. The immunoblots shown do not tell us what cells in those tissues are expressing ESRP1, ESRP2 or both, and when they are expressed. These data are needed to interpret some of these phenotypes, particularly in the more complex genotypes. For example, while reading the manuscript and considering the phenotypes in*
Figure 1
*and*
Figure 2*, the reader is left wondering where ESRP1/2 are expressed. Are the phenotypes developing in bone/limb because their expression is exclusive to these tissues?*

The antibodies used for the Western analysis are not suitable for IF or IHC due to non-specific binding to additional proteins and we have unfortunately been unable to obtain such antibodies through custom projects or screening of commercial antibodies. Hence all histologic or tissue section data on Esrp1 and Esrp2 expression patterns has been at the mRNA level. We previously published comprehensive *in situ* analysis of *Esrp1* expression in whole mouse as well as tissue section in our first report on ESRP1 and ESRP2. In the revision we direct the interested reader to the supplemental figures from this previous study and also summarize relevant organs and tissues that express *Esrp1*. We also highlight results from another group that published *in situ* analysis of *Esrp1* expression in mouse embryos that are also relevant to the pleotropic phenotypes we observed. In the revision we have also added *in situ* analysis (Figure 1—figure supplement 1) for *Esrp2* in whole mouse sections and tissues that shows largely overlapping expression with that for *Esrp1*. We also added data from a previous microarray analysis from numerous mouse cells and tissues to further illustrate the expression patterns for *Esrp1* and *Esrp2* (Figure 1—figure supplement 1). With respect to the limb phenotypes we highlight how *Esrp1* and *Esrp2* expression in ectodermal cells at the apical ectodermal ridge (AER) may impact limb development through regulation of epithelial-mesenchymal crosstalk.

To address the expression of *Esrp1* and *Esrp2* in the cells that comprise the skin and hair follicles we have added *in situ* analysis for *Esrp1* in skin sections showing that it is expressed in all epidermal layers and epithelial cells of the hair follicle (Figure 1—figure supplement 1). This *in situ* analysis also shows that *Esrp1* expression is lost in the KO epidermis. Furthermore, we have included prior published microarray studies from different cell populations in the skin (Figure 1—figure supplement 1). These data also outline the expression domains of *Esrp1* and *Esrp2* in the skin and hair follicle and also demonstrate that they are absent in cells of the dermal papilla and the dermis.

*2)*
Figure 1
*and*
Figure 2
*appear to constitute something of a separate story from the epidermal studies that come later and where the detailed molecular analysis was performed. Since no molecular analysis was performed in the phenotypic tissues of*
Figure 1
*and*
Figure 2*, it is hard to relate these data to the later analysis. The authors could either perform some of these analyses (which may be more difficult in these tissues, moving these data to a supplement, or somehow relating these early figures to a paper that is otherwise about the epidermis*.

We agree that the bulk of the molecular data presented in this manuscript is focused on the epidermis, in part due to the ability to easily purify total epidermis from skin of embryonic mice. However, as the Esrp deficient mice present with a myriad of developmental defects we wanted to highlight several of these key observations in this first publication. The finding of 100% penetrant cleft lip associated with cleft palate (CL/P) as the predominant phenotype in mice with ablation of Esrp1 alone is an interesting finding that we wish to feature in a main figure in this first report on these KO mice. We have therefore kept the Figures showing the craniofacial phenotypes in Figure 1 and added a couple additional panels of data. The limb defects from Figure 1 and the other organogenesis defects we observe in Esrp1/Esrp2 DKO mice from Figure 2 have been moved to Figure 1—figure supplement 3 and Figure 1—figure supplement 4. The identification of the underlying mechanisms for several of these developmental defects are projects currently ongoing, to determine the programs of gene expression and alternative splicing in other epithelial cell populations that relate to these other phenotypes

*3) Wording. The authors use the term “splicing regulatory network” (SRN) to describe their analysis of splicing changes in Esrp1 & Esrp2 KO mice. But the analysis primarily focuses on direct regulatory targets of Esrp1/Esrp2: large changes in the expression of other splicing regulatory factors (SRFs) are not observed, splicing changes in other SRFs were not specifically addressed, nor were other steps taken to infer a network of changes downstream of Esrp perturbation. Therefore, the word “network” does not seem justified and SRN should be replaced by the phrase “splicing regulatory program” or similar*.

We have changed the wording as suggested to describe genome-wide targets of Esrp regulated splicing as a program rather than a network.

*4) The authors describe the GO analysis of the splicing changes, but show no results. They should add this to the supplement and provide p-values (corrected for multiple comparisons)*.

The requested GO analysis corrected for multiple comparisons has been added in Supplemental file 2 and associated changes to the text have been made.

*5) The results would be greatly strengthened by CLIP-seq analysis of an Esrp protein to confirm direct regulatory targets. The authors are likely generating these data and should add it if possible*.

We agree that this is a very important analysis that needs to be carried out and plans are underway to perform CLIP-Seq. Unfortunately, we have been limited by the lack of available antibodies with the specificity needed to IP endogenous Esrp1 to confirm *in vivo* direct targets. To address this shortcoming, we have successfully used CRISPR/Cas9 genome editing to introduce a 2X FLAG tag at the N-terminus of Esrp1 in mouse ES cells and have derived chimeric mice from these cells that should provide us with a tool to carry out CLIP-Seq. However, given the more extended time scale needed to derive these mice with FLAG-tagged Esrp1 this analysis will need to be performed in future investigations.

*6) The authors spend considerable space showing the splicing of FGFR, and discussing this event as an underlying cause of the phenotype, but do not provide evidence. They should be clearer about what portion of the phenotype observed might be attributable to defects in FGF signaling*.

The submitted manuscript did contain some discussions regarding how changes in Fgfr2 splicing may at least partly underlie some of the phenotypes of Esrp1/2 ablation (e.g. lung agenesis). However, we agree that the degree to which alterations in Fgf recptor splicing and Fgf signaling impact the epidermal phenotypes requires further clarification. In the resubmission, we outline in more detail why available evidence suggests that while alterations in FGFR splicing may contribute to the epidermal phenotypes, there are almost certainly roles of other splicing changes in other Esrp target transcripts. The degree to which alterations in Fgfr splicing as well as other splicing events is an topic that will be a focus of further investigations.